# DOT1L inhibition attenuates graft-versus-host disease by allogeneic T cells in adoptive immunotherapy models

Yuki Kagoya[1], Munehide Nakatsugawa[1], Kayoko Saso[1], Tingxi Guo [1,2], Mark Anczurowski[1,2], Chung-Hsi Wang[1,2], Marcus O. Butler [1,2,3], Cheryl H. Arrowsmith[4,5] & Naoto Hirano [1,2]

Adoptive T-cell therapy is a promising therapeutic approach for cancer patients. The use of allogeneic T-cell grafts will improve its applicability and versatility provided that inherent allogeneic responses are controlled. T-cell activation is finely regulated by multiple signaling molecules that are transcriptionally controlled by epigenetic mechanisms. Here we report that inhibiting DOT1L, a histone H3-lysine 79 methyltransferase, alleviates allogeneic T-cell responses. DOT1L inhibition reduces miR-181a expression, which in turn increases the ERK phosphatase DUSP6 expression and selectively ameliorates low-avidity T-cell responses through globally suppressing T-cell activation-induced gene expression alterations. The inhibition of DOT1L or DUSP6 overexpression in T cells attenuates the development of graft-versus-host disease, while retaining potent antitumor activity in xenogeneic and allogeneic adoptive immunotherapy models. These results suggest that DOT1L inhibition may enable the safe and effective use of allogeneic antitumor T cells by suppressing unwanted immunological reactions in adoptive immunotherapy.

[1] Tumor Immunotherapy Program, Campbell Family Institute for Breast Cancer Research, Campbell Family Cancer Research Institute, Princess Margaret Cancer Centre, University Health Network, Toronto, ON M5G 2M9, Canada. [2] Department of Immunology, University of Toronto, Toronto, ON M5S 1A8, Canada. [3] Department of Medicine, University of Toronto, Toronto, ON M5S 1A8, Canada. [4] Structural Genomics Consortium and Department of Medical Biophysics, University of Toronto, Toronto, ON M5G 1L7, Canada. [5] Princess Margaret Cancer Centre, University Health Network, Toronto, ON M5G 2M9, Canada. Correspondence and requests for materials should be addressed to N.H. (email: naoto.hirano@uhnresearch.ca)

T-cell immunotherapy is a potentially curative therapeutic approach for patients with advanced cancer. In addition to the established clinical efficacy of allogeneic stem cell transplantation (allo-SCT) for hematologic malignancies, recent clinical trials have demonstrated that adoptive T-cell therapy (ACT), in which antitumor T cells are expanded in vitro and infused into a patient, induces robust therapeutic effects in some types of cancers[1–5]. Although, autologous T cells have been used as antitumor T-cell grafts in most ACT clinical trials, allogeneic T cells can also serve as an alternative source. Allogeneic viral antigen-specific T cells have successfully been used in the treatment of Epstein–Barr virus-associated posttransplant lympho-proliferative disease, cytomegalovirus, and adenovirus infections[6–9]. Similar attempts have been conducted in the treatment of cancer. Patients undergo allo-SCT from an human leukocyte antigen (HLA)-matched donor or cord-blood cells, followed by infusion of the donor-derived antitumor T-cell grafts[10–13]. Preclinical mouse studies have revealed that the adoptive transfer of allogeneic T cells following lymphodepleting regimens showed therapeutic effects before T-cell rejection[14–16].

Considering the economic constraints imposed by the individual preparation of autologous antitumor T-cell grafts, the standardized production of allogeneic T-cell grafts available for a broad range of patients may provide a more feasible alternative for the widespread clinical application of ACT. However, allogeneic T cells also target normal tissues as allogeneic T-cell responses, which clinically manifests as graft-versus-host disease (GVHD)[17,18]. Although, systemic immunosuppressive therapies have been used for preventing or alleviating GVHD reactions, they are nonselective and inevitably interfere with the desired antitumor immunity. The specific inhibition of GVHD will enable the safe and effective use of allogeneic T-cell grafts.

The interaction between the T-cell receptor (TCR) and peptide-MHC complex (pMHC) triggers a series of intracellular signaling cascades leading to the changes in gene expression programs. These programs are subject to epigenetic regulation at both the transcriptional and posttranscriptional levels. Recent studies have identified a close relationship between epigenetic landscapes and gene expression profiles and functional properties in T cells[19–23]. However, it has yet to be fully elucidated whether the exogenous manipulation of the epigenome can tune TCR signaling. In this study, we extensively explored an epigenetic target that modulates TCR signaling associated with allogeneic T-cell responses. We found that inhibiting DOT1L, a histone H3-lysine 79 methyltransferase, with the specific inhibitor SGC0946 significantly suppressed unwanted allogeneic T-cell responses, while preserving the beneficial antitumor responses in vitro and in multiple adoptive immunotherapy models.

## Results

**DOT1L inhibition represses allogeneic T-cell responses**. To identify an epigenetic target regulating allogeneic T-cell responses, we performed a screening experiment using chemical probes with defined epigenetic modulator and effector protein targets (Supplementary Table 1)[24]. Human CD3$^+$ T cells were stimulated with artificial antigen-presenting cells (aAPCs) that express a membrane-bound form of anti-CD3 monoclonal antibody and the immunostimulatory molecules CD80 and CD83 (aAPC/mOKT3)[25] and were individually treated with various chemical probes. The probe-treated T cells were rested, labeled with carboxyfluorescein succinimidyl ester (CFSE), and then cocultured with peripheral blood mononuclear cells (PBMC) depleted of CD3$^+$ T cells from different donors (Fig. 1a). Allogeneic T-cell responses were evaluated by the induced expression of the activation marker CD69 and CFSE dilution. Among the tested probes, SGC0946, a specific inhibitor of the histone methyltransferase DOT1L, inhibited allogeneic T-cell responses most efficiently (Fig. 1b, c; Supplementary Fig. 1; Supplementary Fig. 2)[26]. DOT1L inhibition did not affect T-cell proliferation in the presence of IL-2 and IL-15, indicating that the attenuated proliferative response against allogeneic PBMC is not due to drug toxicity (Supplementary Fig. 3a, b). Treating T cells with 0.5 μM

SGC0946 progressively reduced histone dimethylation on lysine 79 (H3K79me2) (Supplementary Fig. 4). The suppressive effect of DOT1L inhibition on allogeneic T-cell responses was confirmed with different donor samples (Fig. 1d, e; Supplementary Fig. 5a, b). In addition, the shRNA-mediated knockdown of DOT1L similarly attenuated CD69 upregulation in response to allogeneic PBMC (Fig. 1f; Supplementary Fig. 6a–c). Based on these results, we studied the effect of DOT1L inhibition as a potential strategy to prevent GVHD.

**DOT1L inhibition elevates the TCR stimulation threshold**. We first studied the effect of DOT1L inhibition on maximally stimulated T cells. In contrast to the attenuated allogeneic responses, SGC0946 treatment did not compromise T-cell proliferation, CD69 upregulation, or IFN-γ secretion upon aAPC/mOKT3-based strong stimulation (Fig. 2a–c; the expression profiles of surface molecules in aAPC/mOKT3 are presented in Supplementary Fig. 7a). We also tested the effect of SGC0946 treatment on the cytokine secretion by T cells in response to peptide/MHC complexes. As a model for high-avidity TCR–pMHC interactions, CD3$^+$ T cells were transduced with the HLA-A2/MART1$_{27–35}$-specific TCR clone DMF5 and stimulated with aAPCs expressing HLA-A2 (aAPC/A2) loaded with the A2/MART1$_{27–35}$ peptide (Supplementary Fig. 7b)[27]. DMF5 is a naturally occurring TCR with one of the highest affinities. Similar to stimulation with aAPC/mOKT3, SGC0946-treated CD8$^+$ T cells secreted equivalent levels of IFN-γ compared with control T cells in an A2/MART1-specific manner (Fig. 2d). Allogeneic reactions are supposedly triggered by an array of TCR and allo-pMHC pairs with different affinities. The overall strength of these interactions is considered to be less avid than that in aAPC/mOKT3-mediated or monoclonal high-affinity TCR-mediated reactions. In fact, T-cell stimulation by aAPC/mOKT3 or aAPC/A2 induced significantly higher expression of the activation markers CD25 and CD69, as well as NUR77, an immediate-early response gene induced by TCR engagement, than allogeneic PBMCs (Supplementary Fig. 8a–d). Therefore, we speculated that DOT1L inhibition in T cells elevates the TCR stimulation threshold by dampening low-avidity but not high-avidity T-cell responses. To investigate this possibility, we evaluated the effect of SGC0946 treatment on T cells individually reconstituted with A2/MART1 TCRs with a broad range of affinity that we had previously isolated[28]. Interestingly, SGC0946 treatment did not affect IFN-γ secretion by high-avidity (DMF5) or intermediate-avidity (clone 523 and clone 1086) T cells, but inhibited the secretion by low-avidity T cells (clone 413) (Fig. 2e). In addition, SGC0946 treatment mitigated CD69 induction and cellular division upon antigen-specific stimulation only in the clone 413 TCR-transduced T cells (Fig. 2f, g). Importantly, SGC0946 treatment attenuated the response mediated by high-affinity TCRs upon stimulation with lower concentrations of the cognate peptide (Fig. 2h). We also addressed whether DOT1L inhibition would be

**Fig. 1** Screening of epigenetic targets that regulate allogeneic T-cell responses. **a** Peripheral blood CD3$^+$ T cells were stimulated with artificial antigen-presenting cells that express a membrane-bound form of anti-CD3 mAb (clone OKT3), CD80, and CD83 (aAPC/mOKT3) and individually treated with epigenetic chemical probes for 6 days. The probe-treated T cells were labeled with CFSE and cocultured with T-cell-depleted peripheral blood mononuclear cells (PBMC) from a different donor to evaluate allogeneic T-cell responses or cultured in the presence of IL-2 and IL-15. **b, c** T-cell division (**b**) and expression of CD69 (**c**) in the CFSE-labeled CD4$^+$ and CD8$^+$ T cells were analyzed on day 4 (n = 3 technical replicates, ordinary one-way ANOVA with Tukey's multiple comparisons test). *P < 0.05, **P < 0.01 compared with the DMSO control. Dashed lines denote the average values of the DMSO control samples. **d, e** CD3$^+$ T cells treated with 0.5 μM SGC0946 or DMSO were labeled with CFSE and cocultured with allogeneic PBMC. The frequency of CFSE-diluted cells (**d**) and CD69$^+$ cells (**e**) was analyzed on day 4 (n = 4 cultures, unpaired two-sided t-test). Experiments were repeated three times with different samples, and similar results were obtained. **f** CD3$^+$ T cells were transduced with control shRNA or shRNA against DOT1L and cocultured with allogeneic PBMC. The expression of CD69 in the shRNA-transduced T-cell population was analyzed on day 4 (n = 5 cultures, ordinary one-way ANOVA with Tukey's multiple comparisons test compared with the control). Horizontal lines represent the means ± s.d

useful in modulating TCR cross-reactivity. While control T cells transduced with the high-affinity A2/MART1 TCR clone 830 cross-reacted with peptides derived from the PG transporter, KIAA0735, and the G-protein-coupled receptor RE2 as previously published[28], SGC0946 treatment significantly ameliorated those responses without affecting the response against the MART1 peptide (Fig. 2i). These results suggest that DOT1L inhibition elevates the TCR stimulation threshold of T cells and is potentially applicable in the abrogation of both on-target and off-target low-avidity interactions.

**DOT1L inhibition attenuates xenogeneic GVHD development.** Based on the in vitro data, we investigated whether DOT1L inhibition contributes to the prevention of GVHD in vivo. We first confirmed that it takes ~2 weeks for H3K79me2 repressed by SGC0946 treatment to revert to a normal level of expression upon drug withdrawal (Supplementary Fig. 9a, b). Accordingly, we evaluated the effect of DOT1L inhibition on the development of GVHD by transferring human T cells cultured with or without SGC0946 into irradiated NSG mice (Fig. 3a). DOT1L inhibition did not affect T-cell expansion during in vitro culture

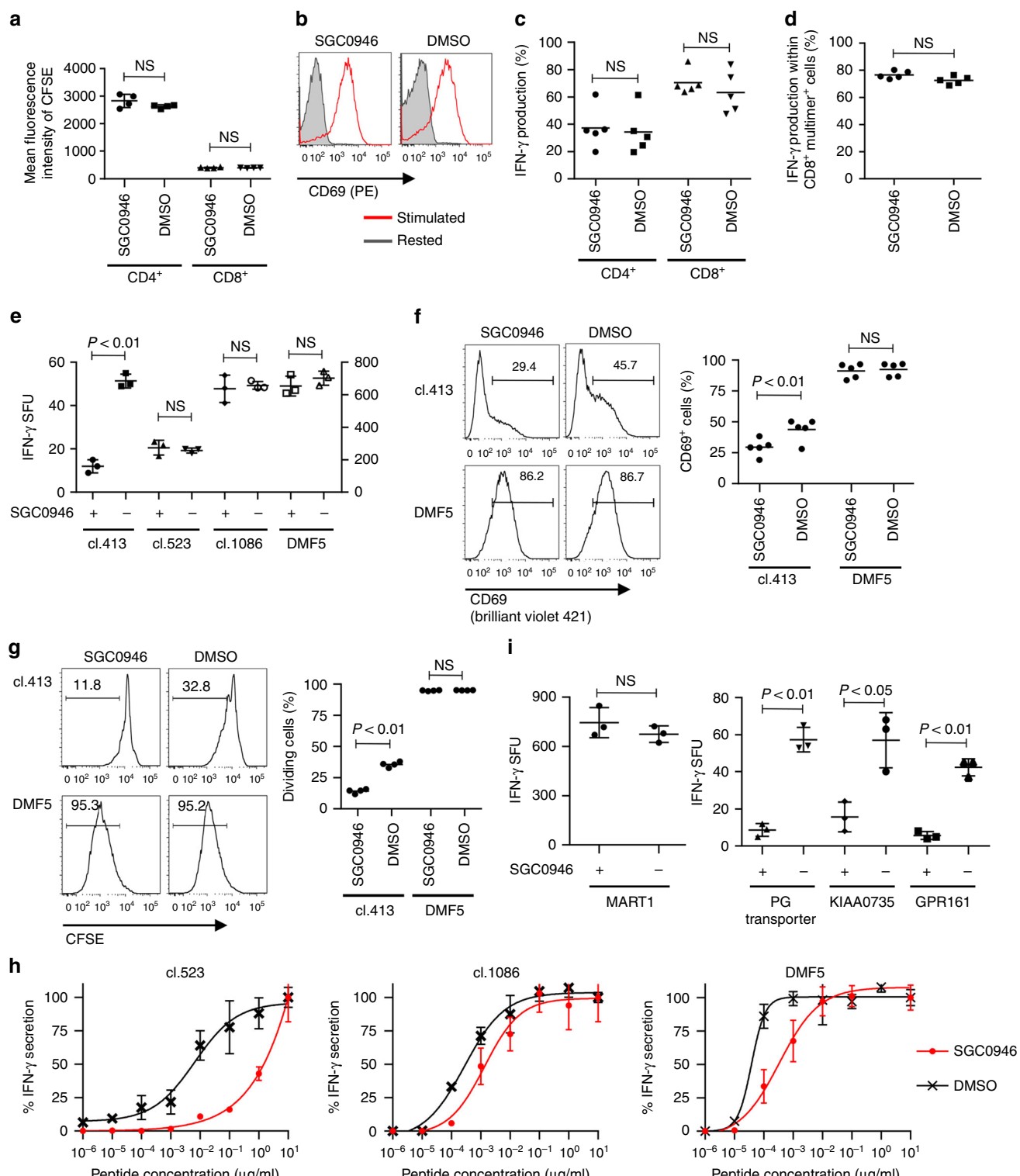

(Supplementary Fig. 10a). Human T cells progressively differentiate from stem cell-like memory T cells ($T_{SCM}$), which have a $CD45RA^+$ $CD62L^+$ $CCR7^+$ phenotype, into central memory T cells ($T_{CM}$, $CD45RA^-$ $CD62L^+$ $CCR7^+$) and then into effector memory T cells ($T_{EM}$, $CD45RA^-$ $CD62L^-$ $CCR7^-$) during in vitro expansion[29,30]. Although, engraftment of human T cells is affected by the differentiation and/or exhaustion status before infusion[24,29], SGC0946 treatment did not significantly alter the expression profiles of memory T-cell markers or exhaustion markers (Supplementary Fig. 10b–e). Moreover, SGC0946-treated T cells maintained their proliferative capacity comparable to control T cells when restimulated by aAPC/mOKT3, suggesting that SGC0946 treatment did not cause significant functional impairment in cultured T cells (Supplementary Fig. 10f).

Intriguingly, while most of the mice injected with the control T cells progressively lost weight in ~2 weeks following adoptive transfer, SGC0946-T cells showed a significantly delayed onset of weight loss, which resulted in a longer overall survival (Fig. 3b–e). GVHD development was also confirmed by histological assessment (Fig. 3f). There was a marked delay in the expansion of human T cells in the peripheral blood and multiple organs (Fig. 3g; Supplementary Fig. 11a). It has recently been reported that GVHD is predominantly induced by $T_{SCM}$. Similar to in vitro culture, these cells undergo proliferation and differentiation into $T_{CM}$ and then into $T_{EM}$ in vivo[24,29]. SGC0946-treated T cells in the peripheral blood maintained a $T_{SCM}$ phenotype within the $CD4^+$ and $CD8^+$ T-cell populations significantly better than the control T cells (Fig. 3h, i). The overall frequency of $T_{SCM}$ in the PBMC did not differ significantly from that of control T cells, suggesting that the lower frequency of SGC0946-treated T cells was not because of poor engraftment but instead resulted from less in vivo expansion of xenogeneic T cells (Fig. 3j). SGC0946-treated T cells also proliferated and underwent differentiation at later time points, which was considered to be due to the recovery of H3K79me2 levels (Supplementary Fig. 11b, c). These results suggest that DOT1L inhibition in human T cells delays the development of GVHD in the in vivo xenogeneic model.

**DOT1L-inhibited T cells show reduced ERK phosphorylation.** To investigate how DOT1L inhibition modulates the T-cell activation threshold, we explored the effect of SGC0946 treatment on the activation of multiple proximal and distal signaling pathways in low-avidity T cells transduced with the A2/MART1 TCR clone 413. Interestingly, phospho-ERK, but not the other phosphoproteins (CD3z, p38 MAPK, and Akt), was significantly downregulated in SGC0946-treated T cells compared with the control (Fig. 4a and Supplementary Fig. 12a). Although, we also analyzed JAK-STAT signaling pathway activation after treatment with cytokines, there were no significant differences in the STAT3 or STAT5 phosphorylation levels between SGC0946- and DMSO-

treated T cells (Supplementary Fig. 12b, c). We then compared whole transcriptomes between SGC0946- and DMSO-treated $CD8^+$ T cells by RNA sequencing analysis. Overall, we extracted 205 genes that were differentially expressed between the DOT1L-inhibited and control T cells ($P < 0.01$, FDR (false discovery rate) <0.1, and fold change of FPKM (fragments per kilobase of transcript per million mapped reads) >1.5 or <0.66 (Supplementary Data 1). Among the gene list, we identified four genes related to the MAPK signaling pathway (Fig. 4b; the MAPK signaling pathway genes were extracted from the KEGG pathway database). Intriguingly, the genes encoding phosphatases (*DUSP2* and *DUSP6*) were significantly upregulated in SGC0946-treated T cells. While DUSP2 is involved in the dephosphorylation of both p38 MAPK and ERK, DUSP6 specifically targets ERK[31,32]. DUSP6 expression was significantly elevated in SGC0946-treated T cells relative to DMSO-T cells at both the mRNA and protein levels (Fig. 4c, d). Since ERK phosphorylation has a critical role in controlling TCR signal strength and activation threshold[33–35], we further examined the effect of DUSP6 upregulation upon DOT1L inhibition. When stimulated with PMA/ionomycin, the peak level of ERK phosphorylation in SGC0946-treated T cells was equivalent to that in control T cells. However, the pERK level decreased more rapidly in SGC0946-treated T cells, corroborating the stronger phosphatase activity in the DOT1L-inhibited cells (Fig. 4e; Supplementary Fig. 13). By the same token, T cells retrovirally transduced with DUSP6 showed more rapid ERK dephosphorylation than the control T cells (Supplementary Fig. 14a–c). The ectopic expression of DUSP6 dampened allogeneic T-cell responses, similar to SGC0946 treatment (Fig. 4f, g). Although, retroviral transduction resulted in significantly higher DUSP6 protein levels than SGC0946 treatment, there were no significant differences in the efficiency of ERK dephosphorylation or suppression of allogeneic responses between DUSP6-transduced and SGC0946-treated T cells. Conversely, ectopic expression of a dominant-negative form of DUSP6/C293S in SGC0946-treated T cells reverted the ERK phosphorylation levels and allogeneic responses to those of the control T cells, suggesting that the increased expression of DUSP6 played a predominant role in the suppression of allogeneic responses in SGC0946-treated T cells (Supplementary Fig. 14d, e). Moreover, the DUSP6-transduced T cells showed less expansion and differentiation than the control T cells after transfer into NSG mice, suggesting that DUSP6 overexpression attenuates xenogeneic T-cell responses in vivo (Fig. 4h, i). Although, we also analyzed the DUSP6 expression levels in T cells treated with other epigenetic chemical probes that modulated allogeneic T-cell responses (Fig. 1), none of the inhibitors affected DUSP6 expression (Supplementary Fig. 15a, b). DUSP6 expression is negatively regulated by microRNA-181a (miR-181a) in thymocytes and peripheral blood T cells[36]. We confirmed that inhibition of miR-

**Fig. 2** DOT1L selectively attenuates low-avidity T-cell responses. **a, b** $CD3^+$ T cells treated with SGC0946 or DMSO were labeled with CFSE and restimulated with aAPC/mOKT3 (day 0). The mean fluorescence intensity of CFSE on day 4 (**a**) and representative CD69 expression on day 1 (**b**) are shown ($n = 4$ cultures, unpaired two-sided $t$-test for each T-cell population). **c** IFN-γ production by SGC0946- or DMSO-treated T cells upon restimulation with aAPC/mOKT3 was analyzed with flow cytometry ($n = 5$ different donor samples, paired two-sided $t$-test). **d** $CD8^+$ T cells retrovirally transduced with HLA-A2/MART1 TCR (clone DMF5) were treated with SGC0946 or DMSO and analyzed for intracellular IFN-γ in the $CD8^+$ $A2/MART1_{27-35}$ multimer$^+$ T-cell population upon stimulation with aAPC/A2 loaded with 10 μg/ml heteroclitic $A2/MART1_{27-35}$ peptide ($n = 5$ different donor samples, paired two-sided $t$-test). **e–h** $CD8^+$ T cells individually transduced with the indicated TCRα/β genes were treated with SGC0946 or DMSO for 1 week and stimulated with aAPC/A2 loaded with 10 μg/ml A2/MART1 peptide. The IFN-γ secretion was analyzed with ELISPOT assays (**e**, $n = 3$ cultures; unpaired two-sided $t$-test). The left $y$-axis is for TCR clone 413, and the right is for the other TCR clones. CD69 expression on day 1 (**f**, $n = 5$ different donor samples, paired two-sided $t$-test) and CFSE dilution on day 4 (**g**, $n = 4$ different donor samples, paired two-sided $t$-test) were analyzed within the TCR-transduced $CD8^+$ T-cell population. The secretion of IFN-γ relative to the maximal response was analyzed by ELISPOT assays (**h**, $n = 3$ cultures for each). **i** A2/MART1 high-avidity T cells (clone 830) were treated with SGC0946 or DMSO and stimulated with aAPC/A2 pulsed with the indicated peptides (10 ng/ml). IFN-γ secretion was analyzed with ELISPOT assays ($n = 3$ cultures, unpaired two-sided $t$-test). NS, not significant. Horizontal lines represent the means ± s.d

181a in cultured T cells elevated DUSP6 expression (Supplementary Fig. 16a, b). SGC0946-treated T cells showed decreased levels of miR-181a relative to control T cells (Fig. 4j). Moreover, treatment with a miR-181a mimic decreased DUSP6 expression in SGC0946-treated T cells to the same level as in control T cells (Fig. 4k, l). SGC0946 treatment reduced the enrichment of H3K79me2 around the transcription start site of miR-181a

(Fig. 4m), suggesting that DOT1L inhibition suppresses miR-181a expression, which results in the increased DUSP6 expression. Although, we further explored several other genes that have previously been reported to regulate DUSP6 expression (*ETS1*, *ETS2*, *PUM2*, and *ZFP36*)[37,38], none of which showed a significant difference in expression levels between control and SGC0946-treated T cells (Supplementary Fig. 17). These results

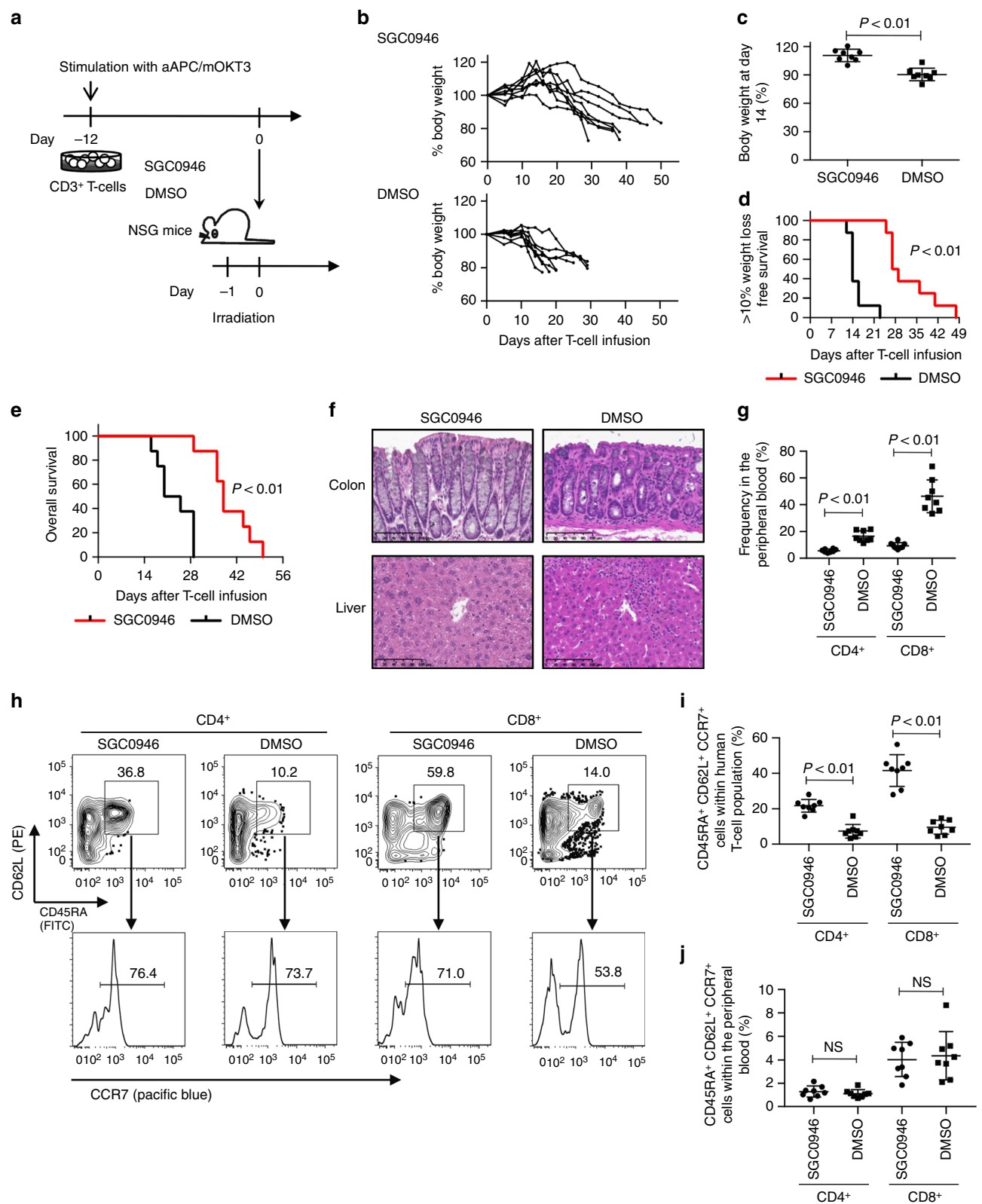

suggest that suppression of miR-181a expression plays a predominant role in DUSP6 upregulation by SGC0946 treatment.

**Gene expression profiles of DOT1L-inhibited T cells.** We further explored how DOT1L inhibition differentially affects high- and low-avidity stimulation-mediated T-cell responses. We analyzed the phosphorylation kinetics of ERK upon stimulation in high- and low-avidity T cells using DMF5- and clone 413-transduced CD8$^+$ T cells. Compared with the DMF5-transduced T cells, the clone 413-transduced T cells showed much weaker increase in the pERK levels (Fig. 5a). While DOT1L inhibition did not significantly alter the initial pERK upregulation in the DMF5-T cells, it dampened pERK levels in the clone 413-transduced T cells throughout the time course analyzed. We then interrogated how the difference in the pERK kinetics affects transcriptomes induced by high-avidity or low-avidity stimulation. CD8$^+$ T cells were transduced with the DMF5 or clone 413 TCR and cultured with or without SGC0946 for 10 days to repress the H3K79me2 levels (Supplementary Fig. 18). The T cells were then stimulated with aAPC/A2 loaded with MART$_{27-35}$ peptide and gene expression profiles of the stimulated T cells were compared.

We extracted 917 genes significantly upregulated or down-regulated upon T-cell activation from the publicly available data (Supplementary Data 2). As expected, unsupervised hierarchical clustering of these genes formed two distinct clusters composed of the DMF5- and clone 413-transduced T cells, respectively (Fig. 5b). Interestingly, SGC0946—and DMSO-treated T cells were grouped within a separate subcluster in the clone 413-T-cell samples, but not in the DMF5-T cells. Principal component analysis also suggested that SGC0946 treatment induced more robust shifts in gene expression profiles in the clone 413-T cells than in the DMF5-T cells (Fig. 5c). When the expression levels of the individual stimulation-associated genes were summed to calculate a T-cell activation score, SGC0946 significantly attenuated stimulation-induced gene expression signature only in the clone 413-T cells, but not in the DMF5-T cells (Fig. 5d). Furthermore, gene set enrichment analysis (GSEA) demonstrated that the clone 413-T cells treated with SGC0946 were negatively enriched with the genes upregulated by T-cell stimulation, and, conversely, positively enriched with the genes downregulated upon stimulation compared with the control clone 413-T cells (Fig. 5e). These genes included activation markers, cell cycle regulators and transcription factors controlling effector differentiation (Supplementary Fig. 19). There was no significant difference in the enrichment of these genes expression profiles between the SGC0946-treated and control DMF5-T cells. These results corroborate that DOT1L inhibition selectively ameliorates low-avidity stimulation-mediated gene expression alterations, which is considered to result in the observed functional differences.

As a comparison, we tested the effect of the MEK inhibitor U0126, which has been reported to ameliorate allogeneic T-cell responses[39,40]. In contrast to SGC0946, treating T cells with U0126 suppressed the initial phase of pERK induction in high-avidity T cells (Supplementary Fig. 20a, b). U0126 treatment suppressed cytokine secretion induced by both low-avidity and high-avidity T-cell stimulation (Fig. 5f). These results suggest that DOT1L inhibition and MEK inhibition modulate T-cell responses through different modes of action.

**DUSP6 overexpression in antitumor T cells ameliorates GVHD.** Finally, we investigated the impact of DUSP6 over-expression on antitumor T-cell responses and the development of GVHD. Human CD3$^+$ T cells were transduced with an anti-CD19 chimeric antigen receptor (CAR) that had CD28 and 4-1BB cytoplasmic domains (28-BB-z)[41]. The CAR gene was linked with truncated nerve growth factor receptor (ΔNGFR) or ΔNGFR and DUSP6 via a P2A sequence. Similar to the data for high-affinity TCR- or anti-CD3 mAb-mediated T-cell stimulation, T-cell stimulation through CAR induced stronger T-cell activation compared with allogeneic reactions (Supplementary Fig. 21a, b). Ectopic expression of DUSP6 did not compromise cytokine secretion or the cytolytic activity of CAR-T cells against the CD19$^+$ acute lymphoblastic leukemia cell line NALM6 in vitro (Fig. 6a, b). Next, we investigated the in vivo attributes of CAR-T cells with or without overexpression of DUSP6. When CFSE-labeled CAR-T cells were transplanted into NALM6-bearing or tumor-free NSG mice, both control and DUSP6-CAR$^+$ T cells proliferated with similar efficiencies in an antigen-dependent manner (Supplementary Fig. 22a–c). There was no significant difference in IFN-γ secretion between control and DUSP6-CAR-T cells in vivo (Supplementary Fig. 22d). To compare the in vivo antileukemic effects of CAR-T cells with or without DUSP6 over-expression, NSG mice that had been intravenously injected with NALM6 expressing EGFP-firefly luciferase fusion gene (NALM6-GL) were treated with the control or DUSP6-expressing CAR-T cells (Fig. 6c; Supplementary Fig. 23a). Longitudinal monitoring of the leukemia cells with bioluminescence imaging showed that both CAR-T cells effectively eradicated NALM6-GL (Fig. 6d; Supplemental Fig. 23b). However, the control CAR-T cells gradually expanded after leukemia cell eradication, which resulted in progressive weight loss and lethality in the transplanted mice (Fig. 6e–h; Supplementary Fig. 23c, d). In contrast, the mice infused with the DUSP6-CAR-T cells survived without leukemia relapse or weight loss. Importantly, DUSP6-CAR-T cells showed similar persistence compared with control CAR-T cells in mice rechallenged with NALM6-GL and prevented leukemia progression (Supplementary Fig. 24a–d). These results suggest that DUSP6 overexpression in CAR-T cells does not impair in vivo CAR-T-cell proliferation in the presence of the target antigen. Similar results were found when T cells were transduced with a different design of the CAR construct that had a 4-1BB cytoplasmic domain alone (BB-z). While both control and DUSP6-overexpressing BB-z CAR-T cells controlled leukemia

**Fig. 3** DOT1L inhibition delays the incidence of GVHD in a xenogeneic model. **a** CD3$^+$ T cells were stimulated with aAPC/mOKT3, expanded for 12 days in the presence of 0.5 μM SGC0946 or DMSO, and subsequently infused into irradiated NSG mice (10 million T cells per mouse). **b**, **c** Relative body weight of the mouse compared to the weight prior to T-cell infusion. Sequential data at the indicated time points (**b**) and the body weight at day 14 (**c**) are shown (n = 8 mice, unpaired two-sided t-test). **d**, **e** Kaplan–Meier analysis for more than 10% weight-loss-free survival (**d**) and overall survival (**e**) following T-cell transplantation (n = 8 mice, log-rank test). The data shown are representative of three independent experiments. **f** Histological assessment of the indicated organs showing lymphocyte infiltration (hematoxylin and eosin staining) on day 21 following T-cell transplantation. Scale bar, 100 μm. **g** Human CD4$^+$ or CD8$^+$ T-cell chimerism in the peripheral blood 14 days following T-cell transfer (n = 8 mice, unpaired two-sided t-test). **h–j** Surface expression of CD45RA, CD62L and CCR7 was analyzed in the persistent T cells within the peripheral blood on day 14. Representative FACS plots (**h**) and frequency of CD45RA$^+$ CD62L$^+$ CCR7$^+$ cells within the CD4$^+$ or CD8$^+$ T-cell population (**i**) or within the total peripheral blood mononuclear cell population (**j**) are shown (n = 8 mice, unpaired two-sided t-test). NS, not significant. Horizontal lines represent the means ± s.d

progression, only control CAR-T cells induced progressive weight loss in treated mice (Supplementary Fig. 25a–f).

To confirm that the expansion of CAR-T cells and development of xenogeneic GVHD after leukemia eradication was induced by endogenous TCR expressed in CAR-T cells, we ablated TCR expression by targeting the TCR alpha constant region with the CRISPR/Cas9 system. The knockout efficiency

was ~30% in both control and DUSP6-CAR-T cells (Supplementary Fig. 26a). Control EGFP[+] CAR-T and DUSP6-ΔNGFR[+] CAR-T cells with or without expression of CD3 were coinfused into NALM6-bearing mice (Fig. 6i). We confirmed the comparable engraftment of each CAR-T-cell population two days after transplantation (Fig. 6j; Supplementary Fig. 26b). However, only EGFP[+] TCR[+] CAR-T cells showed expansion at later time points

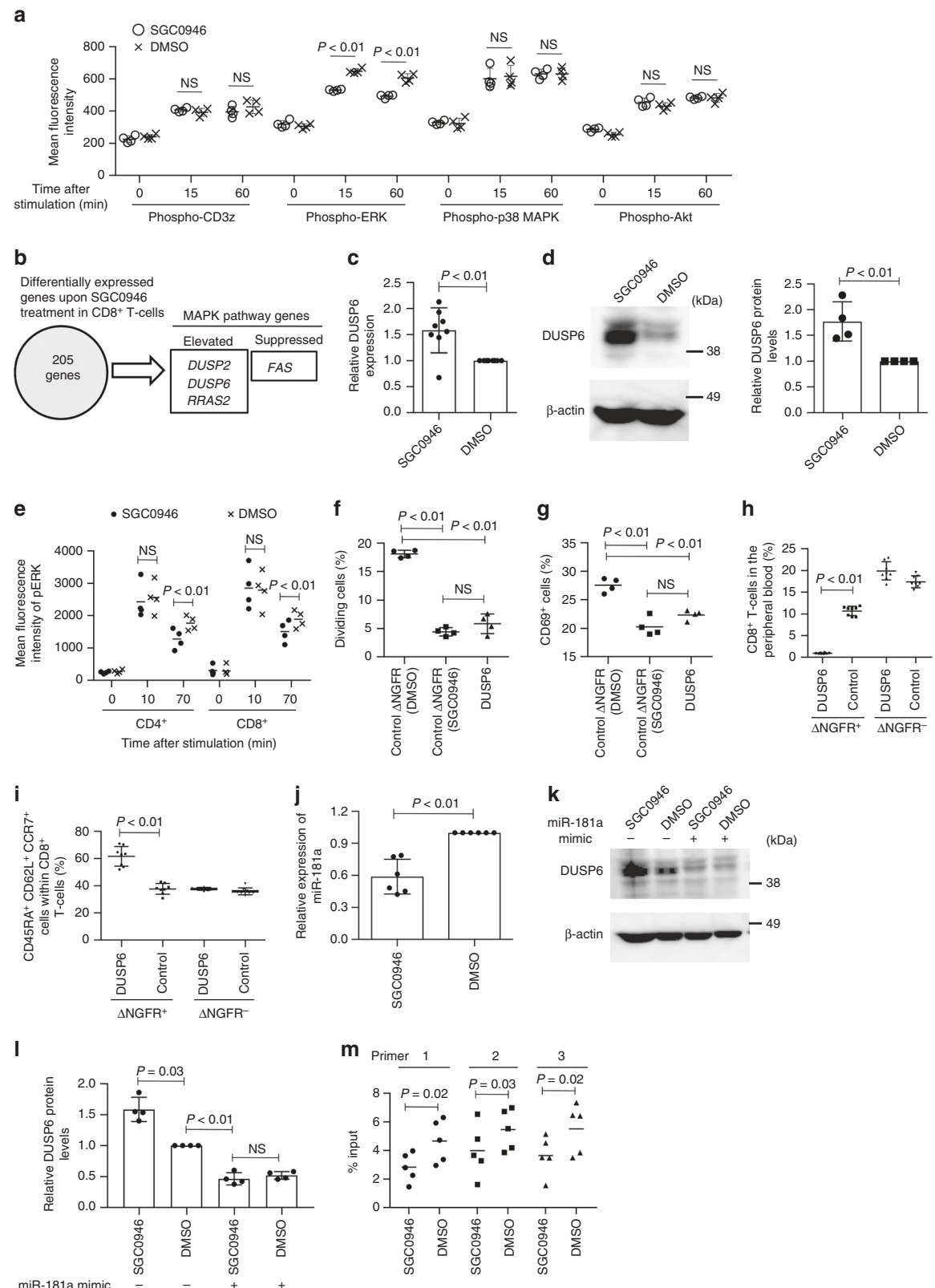

(Fig. 6k). These results suggest that endogenous TCR signaling is required for CAR-T-cell expansion in tumor-eradicated NSG mice through xenogeneic T-cell responses, which is attenuated by overexpression of DUSP6.

**DUSP6 overexpression in allogeneic T cells attenuates GVHD**. We further assessed the impact of DUSP6 expression on allogeneic GVHD and graft-versus-tumor (GVT) effects using an allogeneic mouse model. C57BL/6 mouse-derived T cells showed better in vitro proliferation when cultured with Balb/c mouse-derived A20 lymphoma cells than with normal Balb/c splenocytes or Balb/c-derived fibroblasts, suggesting that stronger allogeneic responses were induced by lymphoma cells (Supplementary Fig. 27a, b). To evaluate the effect of DUSP6 overexpression on the development of GVHD and GVT in vivo, C57BL/6 mouse-derived $CD3^+$ T cells were retrovirally transduced with DUSP6-Furin-SGSG-P2A-$\Delta$NGFR or control $\Delta$NGFR. T cells were cultured for 4 days with human IL-2 (20 IU/ml) and IL-15 (10 ng/ml), and $1 \times 10^5$ of the isolated $\Delta$NGFR$^+$ T cells were coinfused with A20 cells expressing EGFP-luciferase (A20-GL) into sublethally irradiated Balb/c mice (Fig. 7a; Supplementary Fig. 28a, b). The control T cells induced lethal GVHD in the transplanted mice (Fig. 7b, c). In contrast, allogeneic T cells overexpressing DUSP6 did not undergo efficient expansion after transplantation, and were subsequently rejected following the recovery of donor hematopoiesis in the majority of the mice. Both the control and DUSP6-transduced T cells suppressed the progression of A20-GL, which resulted in a better overall survival of the mice treated with the DUSP6-T cells (Fig. 7d, e; Supplementary Fig. 28c–e). Increasing the dose of transferred T cells ($2.5 \times 10^5$ or $5 \times 10^5$ cells per mouse) resulted in a more rapid development of GVHD in control mice, while DUSP6-expressing T cells did not efficiently expand at these doses (Supplementary Fig. 29a–c). We also infused a mixed population of transduced and untransduced T cells and longitudinally monitored the chimerism of the $\Delta$NGFR$^{+/-}$ donor T-cell population at various time points (Fig. 7f). The frequency of DUSP6-$\Delta$NGFR$^+$ T cells was significantly lower than that of control $\Delta$NGFR$^+$ T cells at different time points, while there was no significant difference in the frequency of the $\Delta$NGFR$^-$ T-cell population (Fig. 7g). In addition, the relative abundance of DUSP6$^+$ T cells within the donor T-cell population progressively declined over time after transplantation (Fig. 7h). DUSP6 overexpression did not substantially affect the CD4/8T cell ratio of donor T cells (Fig. 7i). Although, the majority of mice transplanted with both control and DUSP6-T cells developed progressive weight loss and

died of GVHD (Supplementary Fig. 30a, b), the untransduced ($\Delta$NGFR$^-$) donor T-cell population predominated over DUSP6-overexpressing T cells at the time of death (Fig. 7j). These results suggest that DUSP6 overexpression significantly attenuated allogeneic T-cell proliferation compared with untransduced T cells in vivo. We also performed the experiments under different culture conditions: supplementation with a higher dose of IL-2 (300 IU/ml); long-term T-cell culture (7 days). Similar to the results obtained in the above experiments, the infused T cells still induced lethal GVHD under these conditions, and DUSP6-overexpressing T cells showed significantly less efficient expansion compared with control T cells, suggesting that these culture conditions play a minor role in allogeneic T-cell proliferation in vivo (Supplementary Fig. 30c–e). These results collectively indicate that DUSP6-mediated ERK dephosphorylation can segregate GVHD from GVT reactions and may be useful in cancer immunotherapy using allogeneic T cells.

## Discussion

In this study, we identified DOT1L as a novel epigenetic target for regulating the activation threshold in human T cells. Pharmacological or genetic inhibition of DOT1L prevented in vitro allogeneic T-cell responses and xenogeneic GVHD in vivo. Importantly, DOT1L inhibition did not affect high-avidity anti-tumor T-cell responses mediated by the TCR or CAR. The effect of DOT1L inhibition was at least partly through the downregulation of miR-181a, which resulted in the enhanced expression of DUSP6 and promotion of ERK dephosphorylation. Although, the expression of miR-181a is high in early stages of T-cell development and decreases along with T-cell maturation, its expression level is linked to TCR sensitivity in peripheral T cells through modulating ERK phosphatase activity[36,42–44]. Despite the negative impact of DUSP6 on T-cell effector functions, DOT1L inhibition did not affect the cytokine secretion, cytotoxicity or proliferation of T cells stimulated by different aAPCs: aAPC/mOKT3 for polyclonal T-cell stimulation, aAPC/A2 loaded with A2/MART1$_{27-35}$ peptide for the A2/MART1 TCR DMF5-transduced T cells, and K562-CD19 for anti-CD19 CAR-T-cell stimulation. These aAPCs provided substantially stronger activation signals than allogeneic PBMCs, as demonstrated by the increased upregulation of T-cell activation markers in vitro. Similar to our results, Valenzuela et al.[45] reported that PKCθ is required for low-affinity TCR-mediated reactions but is dispensable for those mediated by high-affinity TCRs. Since multiple molecules are involved in TCR stimulation-mediated signal

**Fig. 4** DOT1L inhibition modulates ERK phosphorylation by downregulating miR-181a and consequently increasing DUSP6 expression. **a** CD8$^+$ T cells transduced with a low-affinity A2/MART1 TCR (clone 413) were treated with SGC0946 or DMSO and stimulated with the aAPC/A2 loaded with A2/MART1$_{27-35}$ peptide. The mean fluorescence intensity (MFI) of the indicated phosphoproteins in the TCR-transduced T cells was analyzed ($n = 4$ cultures, unpaired two-sided $t$-test). **b** Gene expression profiles of CD8$^+$ T cells treated with SGC0946 or DMSO for 14 days were compared by RNA sequencing ($n = 3$). **c, d** DUSP6 expression in SGC0946-treated T cells relative to control T cells was analyzed with qPCR (**c**, $n = 8$; one sample $t$-test) and immunoblotting (**d**, representative blots and quantified protein levels, $n = 4$, one sample $t$-test). **e** The MFI of phospho-ERK in SGC0946- or DMSO-treated T cells after stimulation with PMA/ionomycin ($n = 4$ different donor samples, paired two-sided $t$-test). **f, g** CD3$^+$ T cells were transduced with DUSP6-IRES-$\Delta$NGFR or control $\Delta$NGFR and treated with SGC0946 or DMSO. T cells were labeled with CFSE and cocultured with allogeneic PBMCs. The frequencies of dividing cells (**f**) and CD69$^+$ cells (**g**) in the $\Delta$NGFR$^+$ CD8$^+$ T-cell population were analyzed ($n = 4$ cultures, ordinary one-way ANOVA with Tukey's multiple comparisons test). **h, i** CD8$^+$ T cells transduced with DUSP6-IRES-$\Delta$NGFR or control $\Delta$NGFR were transplanted into NSG mice. The frequencies of $\Delta$NGFR$^{+/-}$ CD8$^+$ T cells in peripheral blood (**h**) and CD45RA$^+$ CD62L$^+$ CCR7$^+$ cells within the indicated populations (**i**) were analyzed on day 10 ($n = 8$ mice, unpaired two-sided $t$-test). **j** Relative expression of miR-181a in SGC0946-treated T cells compared with control T cells ($n = 6$, one sample $t$-test). **k, l** SGC0946- or DMSO-treated T cells were transfected with a miR-181a mimic. DUSP6 expression was analyzed by immunoblotting. Representative blots (**k**) and quantified protein levels relative to DMSO control samples are shown (**l**, $n = 4$, repeated measures one-way ANOVA with Tukey's multiple comparisons test). **m** Chromatin samples from SGC0946- or DMSO-treated T cells were immunoprecipitated with an anti-H3K79me2 antibody. The enrichment of H3K79me2 around the transcription start site of miR-181a was analyzed with qPCR ($n = 5$, paired two-sided $t$-test). NS, not significant. Horizontal lines indicate the means ± s.d

transduction, partial inhibition of one of the components might be compensated for upon tonic TCR signaling.

DOT1L inhibition is a potential strategy to prevent GVHD without affecting GVT effects in allo-SCT. Although, the overall affinity range of the TCRs responsible for GVHD and GVT has not been extensively compared, genetic knockout of several molecules related to TCR signaling, such as NFAT1/2, PKCα, and PKCθ, ameliorated GVHD but not GVT activity, suggesting that GVT responses might be driven by more avid interactions than those in GVHD in general[45–47]. Our data suggested that, at least in vitro, A20 lymphoma cells triggered stronger allogeneic T-cell responses compared to normal splenocytes and fibroblasts.

Currently available DOT1L inhibitors have a short half-life in plasma and therefore require continuous infusion for in vivo studies[48]. One of the inhibitors, EPZ-5676, has been tested in a phase I study for acute myeloid leukemia patients with rearrangements of the MLL gene in which the inhibitor is administered as a 28-day continuous intravenous infusion (NCT01684150). In adoptive immunotherapy, the persistence of infused antitumor T cells is highly variable between studies. In a recent clinical trial using CD19 CAR-T cells with a 4-1BB costimulatory domain, CAR-T cells were detected in peripheral blood by flow cytometry up to 11 months[4]. As shown in our study, stable transduction of DUSP6 in CAR- or TCR-engineered antitumor T cells is a

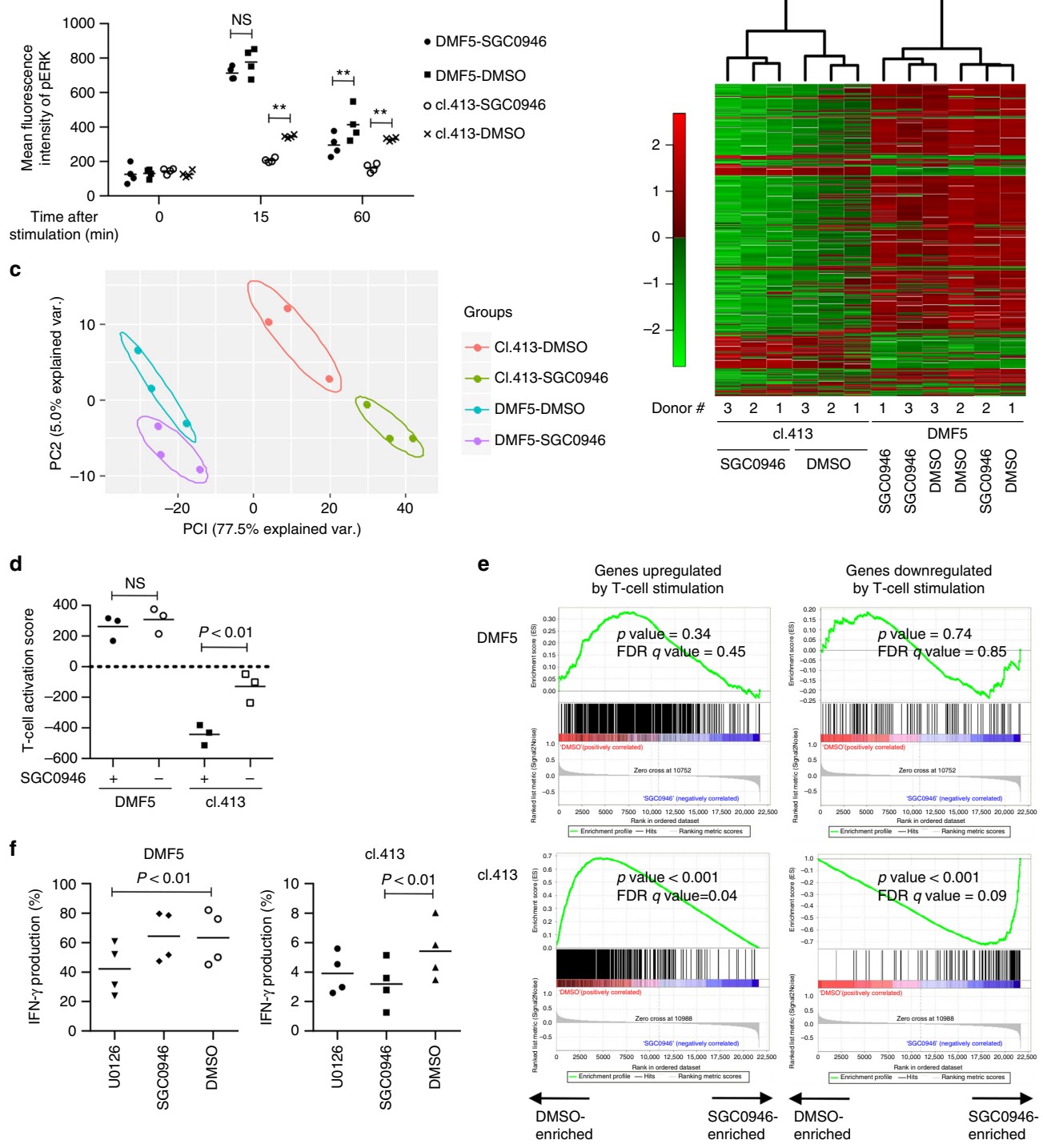

feasible alternative for long-term prevention of GVHD. Ablation of endogenous TCRs using the CRISPR/Cas9 system is another promising approach. However, an optimal protocol to efficiently introduce Cas9/sgRNAs into T cells with minimal toxicity remains to be established. In addition to the prevention of GVHD, our in vitro data suggest that DOT1L inhibition is capable of attenuating T-cell activation by both cognate antigens with low-level expression and cross-reactive antigens with low affinity. These on-target and off-target toxicities have been reported in recent ACT clinical trials; tumor-antigen-specific TCR or CAR-engineered T cells exhibit effector functions against normal tissues that express the target antigens at low levels or different antigens with cross-reactivity[49–53]. Collectively, the present study has identified DOT1L as a novel epigenetic player that modulates the threshold for T-cell activation and could be targeted to prevent a variety of immunological side effects in T-cell immunotherapy.

## Methods

**Reagents.** The epigenetic probes and doses used in the screening of Fig. 1a–c are listed in Supplementary Table 1. The treatment dose was based on the concentrations used in previous studies or by referring to the cell-based assay data available in the SGC (http://www.thesgc.org/chemical-probes/epigenetics). Decitabine and U0126 were purchased from Sigma-Aldrich and Cell Signaling Technologies, respectively. All reagents were dissolved in DMSO except for UNC1215, which was dissolved in water. DMSO was added at a final concentration of 0.01% (volume per volume), including the control samples.

**Cell lines.** Both aAPC/mOKT3 and aAPC/A2 were derived from the human erythroleukemia cell line K562. The aAPC/mOKT3 expresses a membranous form of the anti-CD3 mAb (clone OKT3), as well as the costimulatory molecules CD80 and CD83[25]. The aAPC/A2 expresses HLA-A2, CD80, and CD83[27]. The human acute lymphoblastic leukemia cell line NALM6 was obtained from DSMZ (Braunschweig, Germany). The murine B-cell lymphoma cell line A20 and BALB/3T3 clone A31 were obtained from the American Type Culture Collection (ATCC) (Manassas, VA). All cells were routinely checked for the presence of mycoplasma contamination using PCR-based technology.

**In vitro culture of human and mouse T cells.** Healthy donor-derived PBMC were isolated by Ficoll-Paque PLUS density gradient centrifugation (GE Healthcare). CD3+ or CD8+ T cells were purified through negative magnetic selection using a Pan T Cell Isolation Kit or CD8+ T Cell Isolation Kit (Miltenyi Biotec). Isolated T cells were stimulated with aAPC/mOKT3 irradiated with 120 Gy at an effector to target (E:T) ratio of 10:1. On the following day, 100 IU/ml IL-2 and 10 ng/ml IL-15 (PeproTech) were added to the cultures unless otherwise noted. Culture media were replenished every 3 days.

C57BL/6 mice-derived CD3+ T cells were isolated from the spleen through negative magnetic selection using the Pan T Cell Isolation Kit II, mouse (Miltenyi Biotec). The isolated T cells were stimulated with anti-CD3/CD28 beads (Thermo Fisher Scientific) at a bead to T-cell ratio of 1:2, followed by supplementation with human IL-2 (20 IU/ml) and IL-15 (10 ng/ml) unless otherwise noted.

**Retroviral and lentiviral transduction of T cells.** For lentivirus production, 293T cells were transfected with the pLVX-shRNA2 (Clontech), psPAX2, and pMD2.G plasmids using TransIT-293 (Mirus Bio LLC). Amphotropic retrovirus was produced from PG13 packaging cells stably transduced with each retrovirus plasmid. The virus supernatants were collected 48 h later and immediately used for

transduction. Transduction of human T cells was performed 2 days following stimulation with aAPC/mOKT3 for 3 consecutive days. DUSP6 cDNA was cloned into pMX-internal ribosome entry site-truncated ΔNGFR. A dominant-negative mutant of DUSP6 (C293S) was generated by site-directed mutagenesis[54]. HLA-A2/MART1_{27–35} TCR genes were cloned into the pMX vector. DMF5 was kindly provided by Dr. Rosenberg (NIH/NCI, Bethesda, MD). For the ERK phosphorylation and microarray analysis, these TCR genes were linked to the ΔNGFR gene using a Furin-SGSG-P2A sequence to purify the TCR-transduced T cells through positive magnetic selection of the ΔNGFR+ cells. Anti-CD19 CAR constructs (FMC63 anti-CD19 scFv linked with CD28, 4-1BB and CD3z, or CD8α, 4-1BB, and CD3z) were also inserted into the pMX vector and linked to ΔNGFR or ΔNGFR and DUSP6 by a Furin-SGSG-P2A sequence. The amino acid sequences of the CAR signaling domains are provided in Supplementary Table 2.

Mouse T cells were retrovirally transduced with pMPS-DUSP6-Furin-SGSG-P2A-ΔNGFR or pMPS-ΔNGFR on days 1 and 2 after stimulation. The codon-optimized mouse Dusp6 gene was synthesized by GeneArt (Thermo Fisher Scientific). Phoenix-Eco cells (ATCC) was stably transduced with these plasmids and used for virus production. The produced viruses were coated on a culture plate using Retronectin (Takara Bio).

**Mixed lymphocyte reaction assays.** Cultured T cells were rested overnight in cytokine-free media, labeled with CFSE, and cocultured with allogeneic PBMC at a ratio of 1:2 without cytokine administration. T cells were depleted from the allo-PBMC by positive selection with anti-CD3 MicroBeads (Miltenyi Biotec). CFSE dilution and CD69 expression were analyzed by flow cytometry 4 days following coculture. To analyze allogeneic reactions by murine T cells, C57BL/6 mouse-derived CD3+ T cells were stimulated with anti-CD3/CD28 beads (day 0). On day 4, T cells were labeled with CFSE and cultured with irradiated Balb/c-derived splenocytes, A20 lymphoma cells, or Balb/c-derived 3T3 cells at a ratio of 2:1 without cytokine administration.

**Flow cytometry.** The following antibodies were used for flow cytometry: APC-Cy7-anti-CD4 (clone RPA-T4, BioLegend, #300518; 1:100 dilution), PE-Cy7-anti-CD8 (clone SFCI21Thy2D3, Beckman Coulter, #6603861; 1:200 dilution), PE-Cy5-anti-CD8 (clone B9.11, Beckman Coulter, IM2638U; 1:100 dilution), APC-Cy7-anti-CD3 (clone SK7, BioLegend, #344818; 1:50 dilution), PE-anti-CD69 (clone FN50, BioLegend, #310906; 1:60 dilution), Brilliant Violet 421-antihuman CD69 (clone FN50, BioLegend, #310929; 1:100 dilution), PE-Cy7-anti-CD25 (clone BC96, BioLegend, #302612; 1:60 dilution), FITC-anti-CD45RA (clone MEM-56, Thermo Fisher Scientific, MHCD45RA01; 1:100 dilution), PE-anti-CD62L (clone DREG-56, BioLegend, #304806; 1:25 dilution), Pacific Blue-anti-CCR7 (clone G043H7, BioLegend, #353210; 1:30 dilution), Alexa Fluor 488-anti-CD279 (clone EH12.2H7, BioLegend, #329936; 1:30 dilution), PE-anti-CD274 (clone 29E.2A3, BioLegend, #329706; 1:40 dilution), APC/Cy7-anti-CD366 (clone F38-2E2, Bio-Legend, #345026; 1:40 dilution), PerCP/Cy5.5-anti-CD223 (clone C9B7W, BioLegend, #125212; 1:30 dilution), FITC-anti-CD271 (clone ME20.4, BioLegend, #345104; 1:40 dilution), PerCP/Cy5.5-anti-CD271 (clone ME20.4, BioLegend, #345112; 1:60 dilution), PE-Cy7-anti-CD271 (clone ME20.4, BioLegend, #345109; 1:100 dilution), V450-anti-CD271 (clone C40-1457, BD Biosciences, #562123; 1:100 dilution), APC-anti-CD45 (clone HI30, BioLegend, #304012; 1:60 dilution), PE-anti-CD80 (clone L307.4, BD Biosciences, #340294; 1:20 dilution), PE-anti-CD83 (clone HB15e, BD Biosciences, #550634; 1:20 dilution), FITC-anti-HLA-A2 (clone BB7.2, BioLegend, #343304; 1:30 dilution), PE-anti-mouse IgG (clone Poly4053, BioLegend, #405307, 1:20 dilution), PE-anti-Nur77 (clone D63C5, Cell Signaling Technology, #59999; 1:40 dilution), PE-anti-CD19 (clone HIB19, Bio-Legend, #302208; 1:25 dilution), FITC-anti-mouse CD4 (clone RM4-5, BioLegend, #100510; 1:60 dilution), PE-anti-mouse CD8a (clone 53-6.7, BioLegend, #100708; 1:60 dilution), APC-anti-mouse H-2Kb (clone AF6-88.5, BioLegend, #116518; 1:60 dilution), Alexa Fluor 488-anti-phospho-p44/42 MAPK (clone E10, Cell Signaling Technology, #4374; 1:25 dilution), PE-anti-phospho-p44/42 MAPK (clone 197G2, Cell Signaling Technology, #14095; 1:25 dilution), Alexa Fluor 647-anti-phospho-p44/42 MAPK (Thr202/Tyr204) (clone 20A, BD Biosciences, #561992; 1:20

**Fig. 5** DOT1L inhibition selectively suppresses gene expression alterations induced by low-avidity T-cell stimulation. **a** CD8+ T cells with high avidity (DMF5) and low avidity (clone 413) for A2/MART1 were stimulated with aAPC/A2 loaded with 10 μg/ml heteroclitic A2/MART1_{27–35} peptide with or without SGC0946 treatment. The mean fluorescence intensity of pERK within the CD8+ T-cell population at the indicated time points is shown (n = 4 different donor samples, paired two-sided t-test). **b–e** The DMF5- or clone 413 TCR-transduced CD8+ T cells were treated with SGC0946 or DMSO and stimulated for 24 h with aAPC/A2 loaded with 10 μg/ml heteroclitic A2/MART1_{27–35} peptide. Gene expression profiles of the stimulated T cells were analyzed. Unsupervised hierarchical clustering (**b**) and principal component analysis (**c**) were performed on the genes with altered expression upon TCR stimulation retrieved from GSE13887 (false discovery rate <0.001 and >two-fold change in expression). **d** The T-cell activation score was calculated in each sample using the above gene set (n = 3 different donor samples for each group, repeated measures one-way ANOVA with Tukey's multiple comparisons test). **e** Gene set enrichment analysis (GSEA) on expression profiles of the SGC0946-treated versus DMSO-treated T cells using the genes upregualted or downregualted upon TCR stimulation as gene sets. **f** CD3+ T cells transduced with the A2/MART1 TCR (clone 413 or DMF5) were pretreated with 2.5 μM U0126 or 0.5 μM SGC0946 and stimulated with aAPC/A2 loaded with 10 μg/ml heteroclitic A2/MART1_{27–35} peptide. IFN-γ production normalized to transduction efficiency was analyzed with flow cytometry (n = 4 different donor samples, repeated measures one-way ANOVA with Tukey's multiple comparisons test). **P < 0.01. NS, not significant. Horizontal lines indicate the mean values

dilution), Alexa Fluor 647-anti-phospho-p38 MAPK (Thr180/Thr182) (clone 28B10, Cell Signaling Technology, #4552; 1:25 dilution), anti-phospho-Akt (Thr308) (clone D25E6, Cell Signaling Technology, #13038; 1:100 dilution), Alexa Fluor 647-anti-phospho-STAT3 (Tyr705) (clone 4/P-Stat3, BD Biosciences, #557815; 1:6 dilution), Alexa Fluor 647-anti-phospho-STAT5 (Tyr694) (clone 47/

Stat5, BD Biosciences, #562076; 1:6 dilution), PE-Cy7-anti-IFN-γ (clone 4S.B3, BioLegend, #502528; 1:25 dilution), FITC-anti-IL-2 (clone 5344.111, BD Biosciences, #340448; 1:12.5 dilution), PE-anti-TNF-α (clone MAb11, BioLegend, #502909; 1:50 dilution), and anti-H3K79me2 (clone D15E8, Cell Signaling Technology, #5427; 1:100 dilution). R-PE-anti-rabbit IgG (H + L) (Jackson

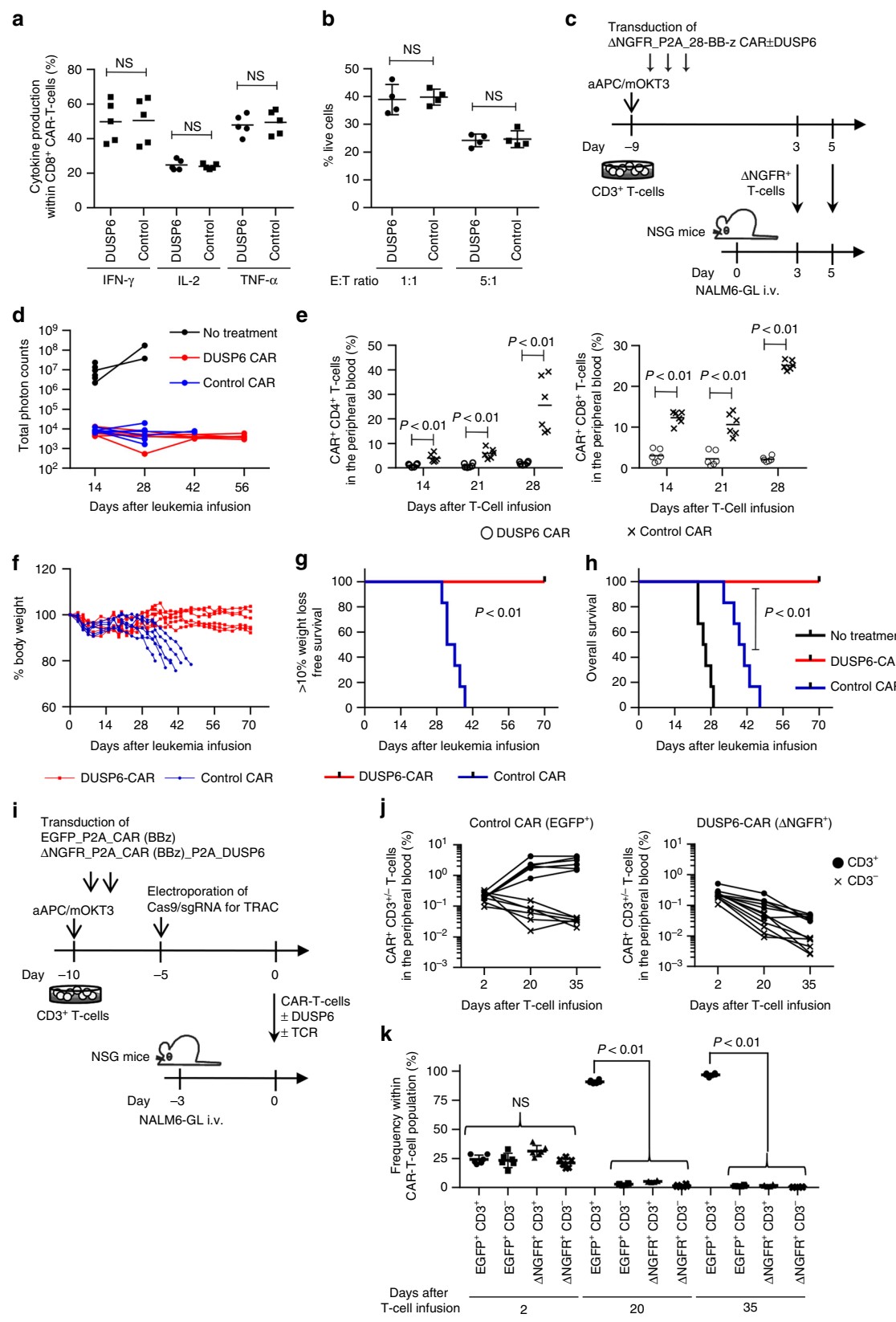

ImmunoResearch, #711-116-152; 1:200 dilution), and Alexa Fluor 647-anti-rabbit IgG (H + L) (Jackson ImmunoResearch, #111-605-144; 1:200 dilution) were used as the secondary antibody following staining with anti-H3K79me2 antibody and anti-phospho-Akt antibody, respectively. The stained cells were analyzed with a FACSCanto II flow cytometer (BD Biosciences). The data analysis were performed with FlowJo software (Tree Star). For the CFSE dilution assay, T cells were labeled with 5 μM CFSE (Thermo Fisher Scientific) before culture. Dead cells were discriminated with the LIVE/DEAD Fixable Dead Cell Stain (Thermo Fisher Scientific, L10119; 1:1000 dilution). Retroviral transduction of the A2/MART1-specific TCR was evaluated using biotinylated HLA-A2/peptide monomers (ProImmune) that were multimerized in-house using streptavidin-PE (Thermo Fisher Scientific) (clone DMF5) or PE-anti-TCR Vbeta 14 (clone CAS1.1.3, Beckman Coulter, IM2047; 1:4 dilution) (clones 413, 523, 830, and 1086) as previously published[28].

**Analysis of cytokine secretion**. For intracellular flow cytometric analysis of cytokine production, T cells were stimulated with aAPC/mOKT3, aAPC/A2 pulsed with heteroclitic MART1$_{27-35}$ peptide, or K562 cells transduced with CD19 at an E:T ratio of 1:1 and incubated for 6 h. Brefeldin A (BioLegend) was added to the cultures 2 h after stimulation. The cells were then fixed and permeabilized using a Cytofix/Cytoperm kit (BD Biosciences). For ELISPOT assays, PVDF plates (Millipore) were coated with antihuman IFN-γ capture mAb (clone 1-D1K, Mabtech, # 3420-3-250; 1:200 dilution). The A2/MART1 TCR-transduced T cells were incubated with aAPC/A2 pulsed with the cognate peptide or the antigen-expressing tumor cells for 24 h at 37 °C. Plates were washed and incubated with biotin-conjugated antihuman IFN-γ detection mAb (clone 7-B6-1, Mabtech, #3420-6-250; 1:2000 dilution) followed by peroxidase-conjugated streptavidin (Jackson ImmunoResearch, #016-030-084; 1:5000 dilution).

**Analysis of phosphoproteins by intracellular flow cytometry**. T cells were rested overnight in cytokine-free media and stimulated on the next day with 100 ng/ml PMA and 1 μg/ml ionomycin, or aAPCs at an E:T ratio of 3:1 for the indicated time periods. The stimulated T cells were first stained for surface molecules, then fixed with 1.8% formaldehyde, permeabilized with ice-cold methanol, and stained with the antibodies against phosphoproteins.

**shRNA interference**. For knockdown experiments, the following sequences were used as targets: CACCTCTGAACTTCAGAAT (sh-*DOT1L*-1) and CCAC-CAACTGCAAACATCA (sh-*DOT1L*-2). The original pLVX-shRNA2 plasmid obtained from Clontech was used as a control. The shRNA-transduced T cells were distinguished as ZsGreen⁺ cells.

**In vitro cytotoxicity assay**. The cytolytic activity of CAR-transduced T cells was analyzed by flow cytometry. Briefly, $1 \times 10^5$ T cells were cocultured with NALM6-GL cells for 4 h at the indicated ratio. The cells were then harvested and stained with a LIVE/DEAD® Fixable Near-IR Dead Cell Stain Kit (Thermo Fisher Scientific). The absolute number of viable GFP⁺ cells was determined with flow cytometry, and the frequency of surviving cells was calculated as a ratio of the cell counts of the samples incubated for the same time period without T cells.

**Mouse experiments**. In the experiments evaluating xenogeneic GVHD in probe-treated human T cells, the 6- to 10-week-old male NSG mice (The Jackson Laboratory) were irradiated (1.5 Gy) and then injected with 10 million cultured CD3⁺ T cells on the following day. The mice were monitored for health and body weight at least three times per week and were killed when they exhibited one of the following symptoms: more than 20% loss of initial body weight, pronounced lethargy, hunched posture, severe diarrhea, or severe dermatitis. DUSP6- or control ΔNGFR-transduced CD8⁺ T cells (3 million cells per mouse) were injected

together with ΔNGFR⁻ CD8⁺ T cells (3 million cells per mouse) and untransduced CD4⁺ T cells (4 million cells per mouse) to elicit engraftment.

In the leukemia treatment model, NALM6 cells were transduced with pMX-EGFP-firefly luciferase (NALM6-GL) for in vivo imaging studies, and $5 \times 10^5$ of these cells were intravenously injected into the irradiated NSG mice. CD3⁺ T cells were retrovirally transduced with each CAR construct and transplanted into the mice. When indicated, the ΔNGFR⁺ cells were magnetically isolated using biotin-conjugated anti-CD271 antibody (clone C40-1457; BD Biosciences, #557195) and anti-biotin MicroBeads (Miltenyi Biotec). Leukemia burden was analyzed with Xenogen IVIS Spectrum (Perkin Elmer). The mice were monitored for health and body weight and were killed when they became moribund or when weight loss exceeded 20% of their body weight before tumor injection. The surviving mice at 70 days after T-cell transplantation were killed and analyzed for the persistence of NALM6-GL and human T cells.

In the experiments to analyze GVHD and GVT in allogeneic mouse model, the 6- to 10-week-old C57BL/6 mouse-derived CD3⁺ T cells were retrovirally transduced with the DUSP6-Furin-SGSG-P2A-ΔNGFR or ΔNGFR, and the indicated numbers of T cells were cotransplanted with A20-GL ($5 \times 10^4$ cells per mouse) into the sublethally irradiated (6 Gy) Balb/c female mice. The mice were monitored for health and body weight every 2–3 days, and were killed when they became moribund or the weight loss exceeded 20% of their body weight before transplantation. The mice surviving on day 30 were killed and analyzed for the persistence of A20-GL and the transplanted T cells within the bone marrow and spleen. The C57BL/6-derived donor T cells were determined as H-2Kb⁺ CD4⁺/CD8⁺ cells. The mice were randomly assigned to treatment groups in each experiment. No mice were excluded from analysis. No statistical methods were used to predetermine sample size. The investigators were not blinded to allocation during experiments and outcome assessment.

**RNA sequencing analysis**. Human CD8⁺ T cells derived from three different donors were stimulated with aAPC/mOKT3 and cultured in the presence of SGC0946 or DMSO for 2 weeks. Total RNA was isolated from each sample by the RNeasy Micro Kit (Qiagen) and used for library preparation. Paired-end RNA sequencing was performed on a HiSeq 2500 (Illumina) by the Centre for Applied Genomics (TCAG) at the Hospital for Sick Children (Toronto, ON). The FASTQ files were trimmed based on a Phred quality score >20 and read lengths >30 using the FASTX Toolkit (version 0.0.14), and the trimmed reads were mapped to the GRCh37 human reference genome using TopHat (version 2.0.14) and Bowtie2 (version 2.2.9). The read counts and FPKM values for each gene were calculated using Cufflinks (v2.1.1). The log$_2$-transformed (FPKM+1) values were compared between SGC0946- and DMSO-treated T cells using the paired $t$-test, and FDR was calculated by the Benjamini–Hochberg method. Differentially expressed genes were extracted based on $P < 0.01$, FDR < 0.1, and a fold change of FPKM values >1.5 or <0.66. Genes with FPKM values <1 in four or more samples were excluded from the analysis.

**Microarray data analysis**. For microarray gene expression analysis of activated T cells, CD8⁺ T cells derived from three different donors were individually transduced with DMF5 or clone 413 TCR and cultured in the presence of SGC0946 or DMSO for 10 days. The probe-treated T cells were then stimulated with aAPC/A2 loaded with 10 μg/ml heteroclitic A2/MART1$_{27-35}$ peptide. RNA was collected 24 h later with RNEasy Micro Kit (Qiagen) and the gene expression profiles were analyzed using the Affymetrix Human Gene 2.0 ST Array by TCAG at the Hospital for Sick Children (Toronto, ON). The data were normalized and annotated to human genes using the Affymetrix Expression Console software version 1.4.1. An unsupervised hierarchical clustering was performed and the heatmap was created using the HeatPlus software package from Bioconductor. A principal component analysis was performed by the function "prcomp" included in the R package "stats" and the data were presented on a two-dimension plot by using the "ggplot2"

**Fig. 6** Elevated DUSP6 expression in chimeric antigen receptor (CAR)-engineered T cells prevents the development of GVHD, while maintaining the antitumor effects. **a**, **b** Human CD3⁺ T cells were transduced with the anti-CD19 CAR linked with ΔNGFR or ΔNGFR and DUSP6. The cytokine production upon stimulation with K562 cells expressing CD19 (**a**, $n = 5$ different donor samples; paired two-sided $t$-test) and cytolytic activity against the CD19⁺ NALM6 leukemia cell line expressing the EGFP-luciferase fusion gene (NALM6-GL) (**b**, $n = 4$ technical replicates; unpaired two-sided $t$-test) by the CAR-T cells were evaluated with flow cytometry. **c** NSG mice were intravenously infused with NALM6-GL (day 0), then with CAR-T cells with or without ectopic expression of DUSP6 on days 3 and 5 (3 million CAR-T cells per infusion). **d** Total photon counts measured by in vivo bioluminescence imaging (IVIS). **e** The frequency of human CD45⁺ CD4⁺/CD8⁺ CAR-T cells in the peripheral blood ($n = 6$ mice for each CAR, unpaired two-sided $t$-test). **f** The sequential monitoring of the body weight relative to the weight on day 0. **g**, **h** Kaplan–Meier analysis for more than 10% weight-loss-free survival (**g**) and overall survival (**h**) after NALM6-GL infusion ($n = 6$ mice for each group, log-rank test). Representative data of two independent experiments. **i**–**k** CD3⁺ T cells were retrovirally transduced with EGFP-P2A-BB-z CAR or ΔNGFR-P2A-BB-z CAR-P2A-DUSP6, and endogenous TCR expression was ablated using CRISPR/Cas9. NALM6-bearing mice were coinfused with CD3⁺/⁻ control and DUSP6-CAR-T cells (4 million T cells per mouse). The persistence of CAR-T cells was monitored in peripheral blood. The frequency of each CAR-T-cell population among peripheral blood mononuclear cells (**j**) or CAR-T cells (**k**) is shown ($n = 6$ mice, repeated measures one-way ANOVA with Tukey's multiple comparisons test). Representative data of two experiments. NS, not significant. Horizontal lines indicate the means ± s.d

package from GitHub. The TCR stimulation-associated genes were retrieved from the publicly available data (GSE13887). Significantly upregulated or downregulated genes (FDR <0.001 and >two-fold change) were selected for the hierarchical clustering and principal component analysis. The T-cell activation score was calculated in individual samples by adding the mean-centered $log_2$-transformed

expression values in the activation-upregulated genes and subtracting the values of the downregulated genes. GSEA was performed with the GSEA v2 software (Broad Institute) using the genes extracted from GSE13887.

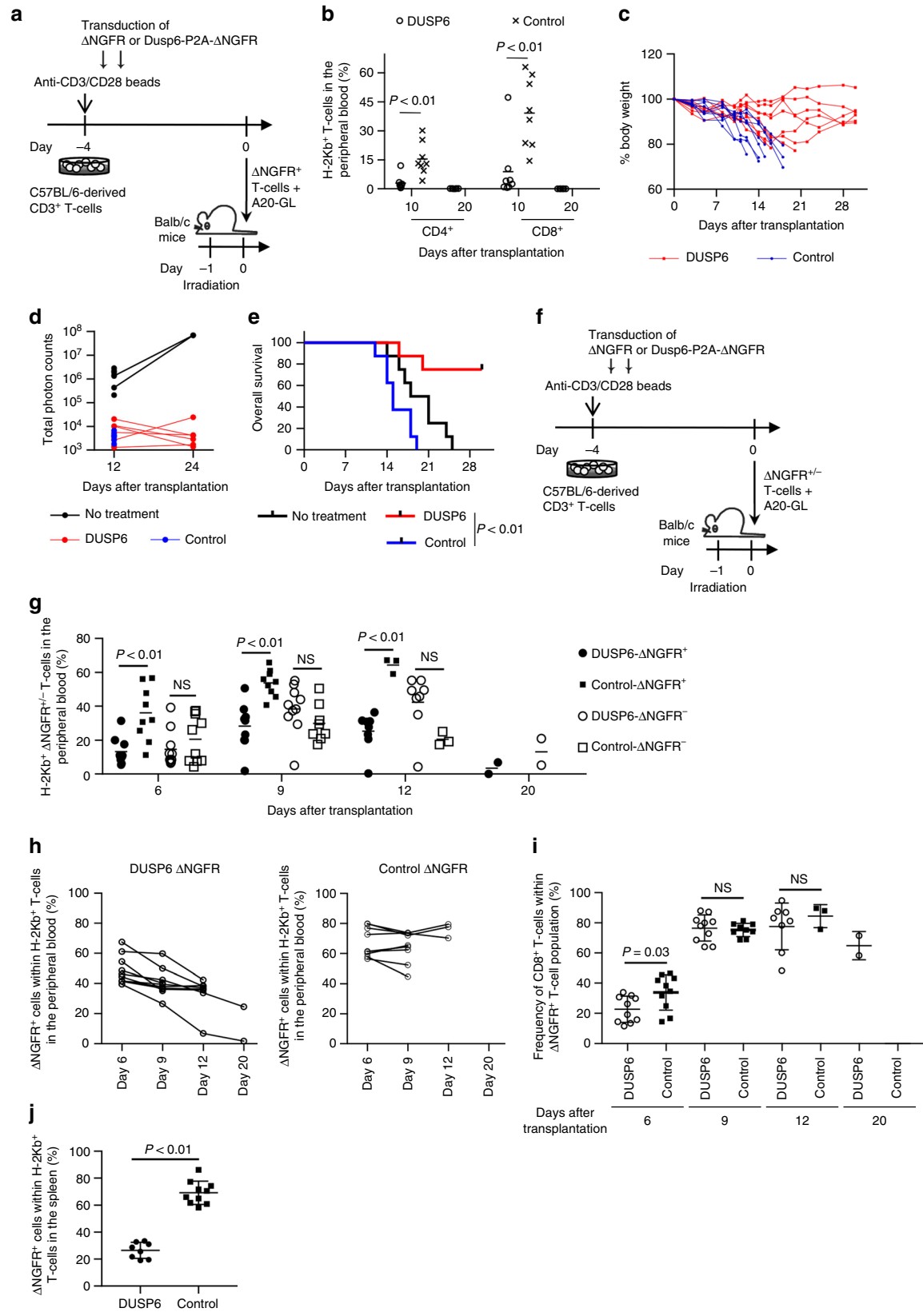

**Fig. 7** DUSP6 overexpression in allogeneic T cells attenuates the development of GVHD without compromising graft-versus-tumor effects. **a–e** C57BL/6 mice-derived CD3$^+$ T cells were transduced with DUSP6-ΔNGFR or ΔNGFR, and the isolated ΔNGFR$^+$ T cells (1 × 10$^5$ T cells per mouse) were transplanted into sublethally irradiated Balb/c mice together with A20 expressing EGFP-luciferase (A20-GL). **b** The frequency of donor T cells in the peripheral blood ($n = 8$ mice for each group, unpaired two-sided $t$-test). **c** The sequential monitoring of the body weight. **d** Total photon counts measured by in vivo bioluminescence imaging. **e** Kaplan–Meier analysis for overall survival (**e**, log-rank test). Representative data of two independent experiments. **f** C57BL/6 mouse-derived CD3$^+$ T cells were transduced with DUSP6-ΔNGFR or ΔNGFR, and a mixed population of ΔNGFR-positive and negative cells (3 × 10$^5$ cells per mouse) and A20-GL were transplanted into sublethally irradiated Balb/c mice. **g** The frequency of ΔNGFR$^{+/-}$ donor T cells was analyzed in peripheral blood (unpaired two-sided $t$-test). **h** The frequency of ΔNGFR$^+$ T cells within the H-2Kb$^+$ donor T-cell population was monitored at the indicated time points. (**i**) The frequency of CD8$^+$ T cells within the ΔNGFR$^+$ donor T-cell population was analyzed (unpaired two-sided $t$-test for each time point). In **g–i**, $n = 9$ for the control-day 9, $n = 3$ for the control-day 12, $n = 8$ for the DUSP6-day 12, $n = 2$ for the DUSP6-day 20, and $n = 10$ for the other plots. **j** At the time of death, the relative abundance of ΔNGFR$^{+/-}$ cells within the H-2Kb$^+$ donor T-cell population was analyzed in the spleen ($n = 8$ for DUSP6-T cells and $n = 10$ for control T cells, unpaired two-sided $t$-test). NS, not significant. Horizontal lines indicate the means ± s.d

**Analysis of serum cytokine concentrations**. The serum concentrations of human IFN-γ in NALM6-bearing or tumor-free NSG mice were measured using an enzyme-linked immunosorbent assay using Human IFN-gamma Quantikine ELISA Kit (R&D Systems) according to the manufacturer's instruction. The concentration was calculated using a four-parameter logistic regression model.

**Immunoblotting**. Equal amounts of protein lysates were separated on 12% (for H3K79me2) or 8% (for the other proteins) gels by SDS-PAGE and transferred to Immobilon-P PVDF membranes (Millipore). The membranes were probed with the primary antibodies at 4 °C overnight, washed and incubated with HRP-conjugated goat anti-mouse IgG (H + L) (Promega, #4021) or anti-rabbit IgG (Promega, #4011) secondary antibody. The following primary antibodies were used: anti-H3K79me2 (clone D15E8, Cell Signaling Technology, #5427; 1:1000 dilution), anti-β-actin (clone C4, Santa Cruz Biotechnology, sc-47778; 1:1000 dilution), and anti-DUSP6 (Cell Signaling Technology, #3058; 1:1000 dilution). Protein levels were quantified with ImageJ software and normalized to β-actin. Uncropped blots are provided in Supplementary Figs. 31–33.

**Quantitative real-time PCR**. RNA was extracted with TRIzol reagent (Thermo Fisher Scientific) and reverse transcribed using Superscript III (Thermo Fisher Scientific). For mature miRNA quantification, the miScript II RT Kit (Qiagen) was used for miRNA reverse transcription. Quantitative real-time PCR was performed on the CFX96 real-time PCR detection system (BioRad) or LightCycler 480 (Roche). SYBR Select Master Mix (Thermo Fisher Scientific) and the miScript SYBR Green PCR Kit (Qiangen) were used to quantify mRNA and miRNA, respectively. The results for mRNA genes were normalized to *UBC*. The expression level of miR-181a was normalized to *U18* expression. Relative expression levels were calculated using the 2-ΔΔCT method. The following primers were used for real-time PCR experiments: *DOT1L* forward, TGGATCACCAGCTGAAGGA, and reverse, AAAGGGTTTCGAGGACACG; *DUSP6* forward, CGACTGGAACGA-GAATACGG, and reverse, GGAGAACTCGGCTTGGAACT; *ETS1* forward, GACCGTGCTGACCTCAATAAG, and reverse, CCATAGCTGGATTGGTCCAC; *ETS2* forward, GAGAGCTTCGAAGATGACTGC, and reverse, CTTGAAAGA-CATGGTTGGCTTA; *PUM2* forward, AGACTGGTGCCTTAGTGGTTG, and reverse, GAGCCATTAACCGAACTGGA; *ZFP36* forward, GTCCTCCAGCTCCTTCTCG, and reverse, GAGGGTGACAGTGGAAGGTC; and *UBC* forward, ATTTGGGTCGCGGGTTCTTG, and reverse, TGCCTTGA-CATTCTCGATGGT. The primers for miR-181a and U18 were included in the miScript primer Assay (Qiagen).

**Chromatin immunoprecipitation**. Cells were crosslinked in 1% formaldehyde for 10 min, and the reaction was quenched with 125 mM glycine for 5 min. Fixed cells were lysed in NP-40 buffer with 0.1% SDS and sonicated with the Sonic Dismembrator Model 100 (Thermo Fisher Scientific). Four microliters of anti-H3K79me2 antibody (clone D15E8, Cell Signaling Technology, #5427) were pre-incubated with Protein-A Dynabeads (Thermo Fisher Scientific) in PBS with 0.5% BSA for 6 h, washed three times and incubated with the lysates overnight. After washing, the samples were eluted in elution buffer (1% SDS, 100 mM NaHCO₃). The eluates were then treated with RNaseA (Thermo Fisher Scientific) for 3 h, and DNA was purified using a MinElute PCR Purification Kit (Qiagen). The collected DNA samples were subjected to qPCR analysis. The data are shown as the ratio of the immunoprecipitated to input samples. The following primers were used for evaluating the enrichment in the region around TSS of miR-181a: primer 1 (−3460 to −3401) forward, TAGTCTTCTGCGGGGTGCTA, and reverse, AAGCTTGGCTCCAATGAGGT; primer 2 (−230 to −156) forward, GCCCAA-TATCGGCCATGTTTT, and reverse, TGGCAAACCTCCACATCACA; and primer 3 (+32 to +129) forward, CGCTGTCGGTGAGTTTGGAA, and reverse, TCAGCGAATTCTGAGCACCA.

**Transfection with a miRNA inhibitor or mimic**. To inhibit or upregulate the function of miR-181a, 2 million T cells were transfected with 200 pmol of the mirVana miR-181a inhibitor, mimic or negative control (Thermo Fisher Scientific) with Nucleofector$^{TM}$ 2b device (Lonza). The T007 program was used for electroporation. Total cell lysates were extracted from transfected T cells after 48 h and used for immunoblotting.

**CRISPR-mediated TCR ablation in CAR-T cells**. Two sgRNAs targeting the TCRα subunit constant gene were generated using the Guide-it sgRNA In Vitro Transcription Kit (Clontech/Takara Bio Inc.) according to the manufacturer's instructions. The target sequences were as follows: 5′-AGAGTCTCTCAGCTGG-TACA-3′ and 5′-TGTGCTAGACATGAGGTCTA-3′. To generate a ribonucleoprotein (RNP) complex, 5 μg of the recombinant *Streptococcus pyogenes* Cas9 protein (SpCas9, Clontech/Takara Bio Inc.) and two sgRNAs (1 μg each) were mixed and incubated at 37° for 5 min. CD3$^+$ T cells were stimulated with aAPC/mOKT3 and retrovirally transduced with a CAR gene on days 2 and 3. On day 4, five million CAR-T cells were electroporated with the RNP complex using the Nucleofector 2b Device and Amaxa Human T cells Nucleofector Kit (Lonza) using the T007 program. The TCR knockout efficiency was evaluated 5 days after electroporation by analyzing CD3 expression.

**Statistics**. The significance of the differences between two groups was assessed with a two-sided paired or unpaired $t$-test. Equality of variance between groups analyzed by an unpaired $t$-test was assessed with an F test. Comparisons between more than two groups were performed by ordinary or repeated measures one-way analysis of variance with Tukey's multiple comparisons tests. The relative mRNA or protein expression levels in the SGC0946-treated cells compared with the control T cells were analyzed with one sample $t$-test. Differences were considered significant at a $P$-value of <0.05. In the mouse GVHD and tumor model experiments, the overall survival and weight loss-free survival of the mice transplanted with T cells were depicted with a Kaplan–Meier curve, and the survival difference between groups was compared with the log-rank test. All statistical analyses were performed using GraphPad Prism 7. Samples sizes were estimated based on preliminary experiments. No statistical method was used to predetermine sample size.

**Study approval**. This study was performed in accordance with the Helsinki Declaration and approved by the Research Ethics Board of the University Health Network, Toronto, Canada. Written informed consent was obtained from all the healthy donors who provided peripheral blood samples. All animal experiments were approved by the Ontario Cancer Institute/Princess Margaret Cancer Centre Animal Care Committee at the University Health Network and performed according to Canadian Council on Animal Care guidelines.

**Data availability**. The microarray and RNA sequencing data have been deposited in the Gene Expression Omnibus (GEO) under the accession number GSE95038 and GSE108694, respectively.

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

## Acknowledgements

This work was supported by Ontario Institute for Cancer Research Clinical Investigator Award IA-039 (N.H.); the Princess Margaret Cancer Foundation (MOB, N.H.); Medicine by Design: A Canada First Research Excellence Fund Program at the University of Toronto (N.H.); the Japan Society for the Promotion of Science Postdoctoral Fellowship for Overseas Researchers (Y.K.); Guglietti Fellowship Award (Y.K.); the Canadian Institutes of Health Research Canada Graduate Scholarship (T.G.); Province of Ontario (T.G., M.A.); and the Natural Sciences and Engineering Research Council of Canada Postgraduate Scholarship (T.G.). The Structural Genomics Consortium (SGC) is a registered charity (number 1097737) that receives funds from AbbVie, Bayer Pharma AG, Boehringer Ingelheim, Canada Foundation for Innovation, Eshelman Institute for Innovation, Genome Canada through Ontario Genomics Institute, Innovative Medicines Initiative (EU/EFPIA) (ULTRA-DD grant no. 115766), Janssen, Merck & Co., Novartis Pharma AG, Ontario Ministry of Economic Development and Innovation, Pfizer, São Paulo Research Foundation-FAPESP, Takeda, and the Wellcome Trust.

## Author contributions

Y.K. and N.H. designed the project. Y.K., M.N., T.G., M.A., C.-H.W., and K.S. performed the experiments. M.O.B. and C.H.A. provided critical reagents and contributed to the writing of this manuscript. Y.K. and N.H. analyzed the results and wrote the manuscript.

## Additional information

**Competing interests:** The authors declare no competing interests.

