## [Peer Review File · Nature Communications]

Reviewers' comments:

Reviewer #1 (Remarks to the Author):

Kagoya et al present an interesting chemical screen for attenuation of T cell activation and proliferation. They identify the histone methyltransferase inhibitor SGC0946 as a small molecule that can attenuate miR181a expression, increase DUSP6 expression and decrease ERK activity. They suggest that SGC0946 may attenuate GVHD without compromising CAR function in primary allogeneic T cells.

The authors present a series of experiments that address the molecular pathways affected by SGC0946, but the functional benefits are not all that convincing as is.

Major points:

1. The authors screen for drugs on aAPCs that presumably provide a strong activating signal (albeit perhaps not; the aAPC used in these studies should be described, including a FACS profile and some estimate of strength of T cell activation). Different activation conditions are used in different experiments (different aAPCs, CD3/CD28 activation, etc). These conditions should be highlighted in discussing different experimental outcomes and summarized in the discussion.
2. The GVHD model is unclear in several aspects. The infusions require a mix of transduced and untransduced T cells (what T cells persist in vivo; what T cells are FACsed at different time points; are the untransduced T cells capable of alloreactivity but inhibited in vivo), only one T cell dose is explored (GVHD may not be prevented at higher T cell doses), the arguments made about low-affinity TCRs are not backed by data or references ("supposedly triggered...", "considered to be..."). It is unclear to what extent T cell culture conditions contribute to the outcome in the model (eg, T cell clearance before any functional recovery).
3. The authors should clearly discuss and discern the effects of SGC0946 on T cell proliferation, differentiation and exhaustion.
4. The CAR studies are unclear and the claims with regards to CARs very broad. The CAR is not described (signaling elements, sequence). It is unclear whether the reported findings would hold up with a different CAR design. The claim that low affinity TCR signaling would be impeded, but not CAR activity, is interesting but unsubstantiated. Clonal studies, perhaps including genetic analyses, would be needed to support this hypothesis. An in vivo experiment demonstrating preserved CAR function while preventing GVHD would provide the most direct functional assessment of this important claim.
5. As the effect of SGC0946 is transient and reversible after 2 weeks, the authors should discuss whether they consider a genetic or chemical approach to be the most clinically relevant.

Reviewer #2 (Remarks to the Author):

Manuscript by Kagoya et al finds that DOT1L inhibition may alleviate allogeneic T-cell response and be useful in preventing GVHD. By screening a library of epigenetic chemical probes, they identified DOT1L inhibitor as one of the most potent agents to alleviate T-cell response. Mechanistic studies linked inhibition of DOT1L with upregulation of DUSP6. Finding potential role of DOT1L inhibition in delaying development of GVHD might have clinical implications. However, further mechanistic studies are needed to explain activity of DOT1L inhibitor.

I have several concerns that should be addressed by additional experiments:

Among epigenetic inhibitors used in the screen, there are several other active compounds (see for example Figure 1c). Are these compounds upregulating DUSP6 or likely work through different mechanism?

It is not clear how DUSP6 was identified as main target gene of DOT1L inhibitor. As described in the text, DUSP6 was selected as one of the genes affected by treatment of MLL-leukemia cell lines with DOT1L inhibitor. This sounds like semi-random guess and it is very likely that expression of many other genes have been affected by treatment with DOT1L inhibitor. More systematic analysis of gene expression by performing RNA-seq is needed in order to better understand the mechanism.

Furthermore, is the DUSP6 protein level achieved upon DUSP6 overexpression in T-cells (experiments in Figure 6) comparable to DUSP6 level upon treatment with DOT1L inhibitor?

Mechanistic link between DOT1L inhibition, downregulation of miR-181a and upregulation of DUSP6 is weak and more data are needed to convincingly validate this mechanism. As presented in Supplementary Fig 4, DOT1L inhibitor completely abolished H3K79me2 and reduction of H3K79me2 level at miR-181a transcription start site is expected. Can you knockdown miR-181a and analyze DUSP6 level? More data are needed to explain how treatment with DOT1L inhibitor upregulates DUSP6.

It is not clear how long was the treatment with DOT1L inhibitor in gene expression studies in activated T-cells (see page 10, Fig 5). Was the treatment long enough to decrease H3K79me2?

Reviewer #1's comments:**Comment #1.**

The authors screen for drugs on aAPCs that presumably provide a strong activating signal (albeit perhaps not; the aAPC used in these studies should be described, including a FACS profile and some estimate of strength of T cell activation). Different activation conditions are used in different experiments (different aAPCs, CD3/CD28 activation, etc). These conditions should be highlighted in discussing different experimental outcomes and summarized in the discussion.

Response to comment #1.

We used two types of aAPCs to stimulate human T cells *in vitro*. For polyclonal T cell stimulation, T cells were cultured with aAPC/mOKT3, which expresses a membranous form of the anti-CD3 mAb (clone OKT3) as well as the costimulatory molecules CD80 and CD83. For antigen-specific stimulation, HLA-A2-MART1 TCR-transduced T cells were cultured with aAPC/A2, which expresses HLA-A2, CD80 and CD83. Both aAPCs are derived from the K562 cell line (Butler et al, Immunol Rev 2014). The expression profiles of surface molecules in these aAPCs are presented in Supplementary Fig. 7, a and b.

To estimate the strength of T cell activation, we stimulated T cells with aAPC/mOKT3, aAPC/A2 loaded with 10 µg/ml MART1₂₇₋₃₅ peptide or allogeneic PBMCs and analyzed the surface expression of CD69 and CD25 and upregulation of NUR77, which is induced in response to TCR stimulation (Liu et al, Nature 1994). For stimulation by aAPC/A2, T cells were retrovirally transduced with the A2/MART1 high-affinity TCR DMF5. As shown in Supplementary Fig. 8, a-d, stimulation by aAPC/mOKT3 and aAPC/A2 induced significantly higher levels of CD69, CD25 and NUR77 expression compared with allogeneic PBMCs.

Anti-CD19 CAR-transduced T cells were stimulated with K562 transduced with CD19 for *in vitro* experiments. We confirmed that K562-CD19 upregulated CD25 and CD69 in CAR-T cells at significantly higher levels than allogeneic PBMCs (Supplementary Fig. 21). These results indicate that stimulation by aAPCs in this study provides stronger T cell activation than allogeneic PBMCs. These data are summarized in the Discussion as shown below.

* Page 6, line 16: **In fact, T cell stimulation by aAPC/mOKT3 or aAPC/A2 induced significantly higher expression of the activation markers CD25 and CD69, as well as NUR77, an immediate-early response gene induced by TCR engagement, than allogeneic PBMCs (Supplementary Fig. 8a-d).**

* Supplementary Fig. 7:

Supplementary Fig. 7. K562-based artificial antigen-presenting cells (aAPCs). (a, b) Surface expression of the indicated molecules in aAPC/mOKT3 (a) and aAPC/A2 (b). Expression of a membranous form of anti-CD3 mAb (clone OKT3) was detected using anti-mouse IgG.

* Supplementary Fig. 8:

Supplementary Fig. 8. Comparison of the strength of T cell activation by aAPCs and allogeneic PBMCs.

(a-d) CD3⁺ T cells were retrovirally transduced with control ΔNGFR or DMF5 TCR-P2A-ΔNGFR and cultured with the indicated cells. Surface expression of CD69 and CD25 (a), or intracellular expression of NUR77 (c) within the CD8⁺ ΔNGFR⁺ T cell population was analyzed on the next day. DMF5-transduced T cells were used for stimulation by the aAPC/A2 loaded with MART1₂₇₋₃₅ peptide, and control ΔNGFR-transduced T cells for the other stimulations. The mean fluorescence intensity of each molecule is shown in (b) and (d) (n=4 technical replicates, ordinary one-way ANOVA with Tukey's multiple comparisons test). Horizontal lines indicate the means ± s.d. Similar results were reproduced in a repeated experiment. ** P<0.01. ns, not significant.

* Page 13, Line 13: Similar to the data for high-affinity TCR- or anti-CD3 mAb-mediated T cell stimulation, T cell stimulation through CAR induced stronger T cell activation compared with allogeneic reactions (Supplementary Fig. 21a,b).

* Supplementary Fig. 21:

Supplementary Fig. 21. T cell activation mediated by a chimeric antigen receptor (CAR). (a-d) CD3⁺ T cells were retrovirally transduced with control Δ NGFR or Δ NGFR-P2A-anti-CD19 CAR with CD28 and 4-1BB domains (28-BB-z) and stimulated with the indicated cells. Surface expression of CD69 and CD25 within the CD8⁺ Δ NGFR⁺ T cell population was analyzed the next day. CAR-transduced T cells were cultured with K562 expressing CD19 (K562-CD19), while control Δ NGFR-transduced T cells were stimulated with aAPC/mOKT3 or allogeneic PBMC. Representative FACS plots (a) and the mean fluorescence intensity for each expression are shown (b, n=4 technical replicates, ordinary one-way ANOVA with Tukey's multiple comparisons test). Horizontal lines indicate the means \pm s.d. Similar results were reproduced in a repeated experiment. ** P<0.01.

* Page 17, line 10: Despite the negative impact of DUSP6 on T cell effector functions, DOT1L inhibition did not affect the cytokine secretion, cytotoxicity or proliferation of T cells stimulated by different aAPCs: aAPC/mOKT3 for polyclonal T cell stimulation, aAPC/A2 loaded with A2/MART1₂₇₋₃₅ peptide for the A2/MART1 TCR DMF5-transduced T cells, and K562-CD19 for anti-CD19 CAR-T cell stimulation. These aAPCs provided substantially stronger activation signals than allogeneic PBMCs, as demonstrated by the increased upregulation of T cell activation markers *in vitro*.

Comment #2-1.

The GVHD model is unclear in several aspects. The infusions require a mix of transduced and untransduced T cells (what T cells persist *in vivo*; what T cells are FACsed at different time points; are the untransduced T cells capable of alloreactivity but inhibited *in vivo*)

Response to comment #2-1.

Thank you for this important comment. In the revised manuscript, we infused a mixed population of transduced and untransduced T cells (3×10^5 cells per mouse) and analyzed the expansion of the transplanted T cells in peripheral blood on days 6, 9, 12, and 20 (Fig. 7f). As shown in Fig. 7, g and h, the frequency of the Δ NGFR⁺ T cell population in peripheral blood was significantly higher at each time point in the control T cell group than the DUSP6-T cell group. Moreover, the relative abundance of Δ NGFR⁺ DUSP6⁺ T cells in the donor T cell population progressively declined over the time course. At the time of death, the donor T cell population was predominantly composed of untransduced (Δ NGFR⁻) T cells in the DUSP6 group, while 60-80% of donor T cells were positive for Δ NGFR expression in the control group (Fig. 7j). Most transplanted mice developed lethal GVHD in both the control and DUSP6-T cell groups, and there was only a slight difference in overall survival between the groups (Supplementary Fig. 30b). These results suggest that untransduced T cells can also expand and induce GVHD and that DUSP6-overexpressing T cells have significantly attenuated allogeneic T cell proliferation *in vivo* compared with untransduced T cells.

* Page 16, line 2: We also infused a mixed population of transduced and untransduced T cells and longitudinally monitored the chimerism of the Δ NGFR^{+/-} donor T cell population at various time

points (Fig. 7f). The frequency of DUSP6- Δ NGFR⁺ T cells was significantly lower than that of control Δ NGFR⁺ T cells at different time points, while there was no significant difference in the frequency of the Δ NGFR⁻ T cell population (Fig. 7g). In addition, the relative abundance of DUSP6⁺ T cells within the donor T cell population progressively declined over time after transplantation (Fig. 7h). DUSP6 overexpression did not substantially affect the CD4/8 T cell ratio of donor T cells (Fig. 7i). Although the majority of mice transplanted with both control and DUSP6-T cells developed progressive weight loss and died of GVHD (Supplementary Fig. 30a,b), the untransduced (Δ NGFR⁻) donor T cell population predominated over DUSP6-overexpressing T cells at the time of death (Fig. 7j). These results suggest that DUSP6 overexpression significantly attenuated allogeneic T cell proliferation compared with untransduced T cells *in vivo*.

* Fig. 7f-j:

Fig. 7. (f) C57BL/6 mouse-derived CD3⁺ T cells were transduced with DUSP6-ΔNGFR or ΔNGFR, and a mixed population of ΔNGFR-positive and negative cells (3×10^5 cells per mouse) and A20-GL were transplanted into sublethally irradiated Balb/c mice. (g) The frequency of ΔNGFR^{+/+} donor T cells was analyzed in peripheral blood (n=10 mice, unpaired *t*-test). (h) The frequency of ΔNGFR⁺ T cells within the H-2Kb⁺ donor T cell population was monitored at the indicated time points. (i) The frequency of CD8⁺ T cells within the ΔNGFR⁺ donor T cell population was analyzed (unpaired *t*-test for each time point). In g-i, n=9 for the control-day 9, n=3 for the control-day 12, n=8 for the DUSP6-day 12, n=2 for the DUSP6-day 20, and n=10 for the other plots. (j) At the time of death, the relative abundance of ΔNGFR^{+/+} cells within the H-2Kb⁺ donor T cell population was analyzed in the spleen (n=8 for DUSP6-T cells and n=10 for control T cells, unpaired *t*-test). ns, not significant. Horizontal lines indicate the means \pm s.d.

* Supplementary Fig. 30a,b:

Supplementary Fig. 30. Comparison of allogeneic T cell proliferation between DUSP6-transduced and untransduced T cells. (a,b) CD3⁺ T cells from C57BL/6 mice were retrovirally transduced with control ΔNGFR, or DUSP6-P2A-ΔNGFR and infused into irradiated Balb/c mice as described in Fig. 7f. Serial monitoring of body weight (a) and overall survival (b) of transplanted mice (n=10 mice for each group, log-rank test).

Comment #2-2.

Only one T cell dose is explored (GVHD may not be prevented at higher T cell doses).

Response to comment #2-2

When the dose of transplanted T cells was increased (2.5×10^5 or 5×10^5 cells), control T cell-transplanted mice developed GVHD more rapidly than mice that received 1×10^5 T cells. By contrast, DUSP6-transduced T cells induced GVHD in only a portion of mice (1/8 mouse in 2.5×10^5 cells and 2/8 mice for 5×10^5 cells). These results suggest that DUSP6-T cells cannot induce lethal GVHD efficiently, even if the dose of infused T cells is increased. Note that Δ NGFR⁺ cells were isolated before transplantation in these experiments because untransduced cells expanded faster than DUSP6-overexpressing T cells, as shown above, making it difficult to evaluate the GVHD-inducing capacity of DUSP6⁺ T cells.

* Page 15, Line 23: Increasing the dose of transferred T cells (2.5×10^5 or 5×10^5 cells / mouse) resulted in a more rapid development of GVHD in control mice, while DUSP6-expressing T cells did not efficiently expand at these doses (Supplementary Fig. 29a-c).

* Supplementary Fig. 29:

Supplementary Fig. 29. Dose escalation of infused allogeneic T cells. (a-c) CD3⁺ T cells from C57BL/6 mice were retrovirally transduced with Δ NGFR, or codon-optimized *Dusp6* and Δ NGFR linked to a Furin-SGSG-P2A sequence. Isolated Δ NGFR⁺ T cells (2.5×10^5 or 5×10^5) together with A20-GL were infused into irradiated Balb/c mice. Serial monitoring of body weight (a), overall survival (b), and frequency of H-2Kb⁺ Δ NGFR⁺ T cells in transplanted mice are shown (n=8 mice for each group; b, log-rank test; c, unpaired *t*-test). The data shown are a composite of three independent experiments. Horizontal lines indicate the mean values.

Commnet #2-3.

The arguments made about low-affinity TCRs are not backed by data or references (“supposedly triggered...”, “considered to be...”).

Response to comment #2-3

We compared the strength of the allogeneic T cell responses triggered by A20 lymphoma cells, Balb/c splenocytes, and Balb/c fibroblasts (3T3 cell line) *in vitro*. CFSE-labelled C57BL/6 T cells were individually cultured with these cells, and T cell proliferation was evaluated by the CFSE dilution. As shown in Supplementary Fig. 27, A20 cells induced T cell proliferation most efficiently, suggesting a stronger allo-T cell response against tumor cells compared with normal cells.

* Page 15, Line 9: C57BL/6 mouse-derived T cells showed better *in vitro* proliferation when cultured with Balb/c mouse-derived A20 lymphoma cells than with normal Balb/c splenocytes or Balb/c-derived fibroblasts, suggesting that stronger allogeneic responses were induced by lymphoma cells (Supplementary Fig. 27a,b).

* Supplementary Fig. 27:

Supplementary Fig. 27. Comparison of the allogeneic T cell responses between tumor cells and normal cells. (a) CD3⁺ T cells derived from C57BL/6 mice were primed with anti-CD3/CD28 beads. On day 4, CFSE-labelled T cells were cultured with A20 lymphoma cells, Balb/c splenocytes or Balb/c-derived 3T3 cells, and the CFSE dilution was analyzed 4 days later. Representative FACS plots (a) and the frequency of CFSE-diluting cells are shown (b, n=4 cultures, ordinary one-way ANOVA with Tukey's multiple comparisons test). Horizontal lines denote the means \pm s.d. ** P<0.01.

Comment #2-4.

It is unclear to what extent T cell culture conditions contribute to the outcome in the model (eg, T cell clearance before any functional recovery).

Response to comment #2-4.

In the original manuscript, T cells were cultured for 4 days in the presence of human IL-2 (20 IU/ml) and IL-15 (10 ng/ml). In the revised manuscript, we tested two additional culture conditions: (i) supplementation with a higher-dose IL-2 (300 IU/ml) without IL-15 and (ii) T cell culture for 7 days. Supplementation with different cytokines significantly affects T cell phenotypes. While IL-2 promotes T cell differentiation towards effector cells, IL-15 helps to maintain less differentiated memory T cells (Manjunath et al, J Clin Invest 2001). In addition, T cells undergo progressive differentiation upon long-term culture (Gattinoni et al, Nat Med 2009).

As shown in Supplementary Fig. 30c-e, these culture conditions did not substantially affect the development of GVHD or overall survival compared with the original culture condition (Supplementary Fig. 30a,b). Δ NGFR⁺ DUSP6-T cells showed less expansion than control Δ NGFR⁺ cells, as observed in the original culture condition (Fig. 7g).

* Page 16, Line 15: We also performed the experiments under different culture conditions: supplementation with a higher dose of IL-2 (300 IU/ml); longer-term T cell culture (seven days). Similar to the results obtained in the above experiments, the infused T cells still induced lethal GVHD under these conditions, and DUSP6-overexpressing T cells showed significantly less efficient expansion compared with control T cells, suggesting that these culture conditions play a minor role in allogeneic T cell proliferation *in vivo* (Supplementary Fig. 30c-e).

* Supplementary Fig. 30c-e:

Supplementary Fig. 30. (c, d) C57BL/6-derived CD3⁺ T cells were transduced with control Δ NGFR, or DUSP6-P2A- Δ NGFR and cultured with 300 IU/ml IL-2 for 4 days or with 20 IU/ml IL-2 and 10 ng/ml IL-15 for 7 days, and 3×10^5 Δ NGFR^{+/-} T cells were transplanted into irradiated Balb/c mice. Serial monitoring of body weight (c) and overall survival (d) of transplanted mice are shown (n=8 mice for each group; d, log-rank test). (e) The frequency of Δ NGFR^{+/-} donor T cells cultured under the indicated conditions was monitored in peripheral blood (n=8, unpaired *t*-test). Horizontal lines indicate the mean values. ns, not significant.

Comment #3.

The authors should clearly discuss and discern the effects of SGC0946 on T cell proliferation, differentiation and exhaustion.

Response to comment #3

We evaluated the effects of SGC0946 treatment on T cell proliferation, differentiation and exhaustion *in vitro*. CD3⁺ T cells were stimulated with aAPC/mOKT3 and cultured in the presence of SGC0946 (0.5 μM) or DMSO for 12 days.

SGC0946 treatment did not affect T cell proliferation in multiple donor samples (Supplementary Fig. 10a). On day 12, we analyzed T cell differentiation by expression of CD45RA, CD62L and CCR7 in the CD4⁺ and CD8⁺ T cell populations. There was no significant difference in the frequency of the CD45RA⁺ CD62L⁺ CCR7⁺ (stem cell-like memory), CD45RA⁻ CD62L⁺ CCR7⁺ (central memory), or CD45RA⁻ CD62L⁻ CCR7⁻ (effector memory) T cell population between SGC0946- and DMSO-treated T cells (Supplementary Fig. 10b,c). We also confirmed that SGC0946 did not alter the surface expression of exhaustion markers: PD-1, PD-L1, LAG-3 and TIM-3 (Supplementary Fig. 10d,e). Impaired proliferation upon restimulation is a hallmark of T cell exhaustion. SGC0946- or DMSO-treated T cells showed comparable proliferation upon restimulation with aAPC/mOKT3 on day 12, suggesting that SGC0946 treatment did not impair the proliferative capacity of T cells (Supplementary Fig. 10f).

* Page 7, Line 21: DOT1L inhibition did not affect T cell expansion during *in vitro* culture (Supplementary Fig. 10a). Human T cells progressively differentiate from stem cell-like memory T cells (T_{SCM}), which have a CD45RA⁺ CD62L⁺ CCR7⁺ phenotype, into central memory T cells (T_{CM}, CD45RA⁻ CD62L⁺ CCR7⁺) and then into effector memory T cells (T_{EM}, CD45RA⁻ CD62L⁻ CCR7⁻) during *in vitro* expansion^{29,30}. Although engraftment of human T cells is affected by the differentiation and/or exhaustion status before infusion^{24,29}, SGC0946 treatment did not significantly alter the expression profiles of memory T cell markers or exhaustion markers (Supplementary Fig. 10b-e). Moreover, SGC0946-treated T cells maintained their proliferative capacity comparable to control T cells when restimulated by aAPC/mOKT3, suggesting that SGC0946 treatment did not cause significant functional impairment in cultured T cells (Supplementary Fig. 10f).

* Supplementary Fig. 10:

Supplementary Fig. 10. Effects of SGC0946 treatment on T cell proliferation and differentiation. (a) CD3⁺ T cells were stimulated with aAPC/mOKT3 (day 0). On the next day, T cells were split into two wells and cultured in the presence of IL-2 (100 IU/ml), IL-15 (10 ng/ml), and SGC0946 or DMSO. The fold expansion of T cells at each time point was calculated (n=5 different donor samples, paired *t*-test). (b, c) Surface expression of CD45RA, CD62L and CCR7 was analyzed on day 12. Representative FACS plots (b) and the frequencies of CD45RA⁺ CD62L⁺ CCR7⁺, CD45RA⁻ CD62L⁺ CCR7⁺, and CD45RA⁻ CD62L⁻ CCR7⁻ cells within the CD4⁺ or CD8⁺ T cell population (c) are shown (n=9 different donor samples, paired *t*-test for each population). (d, e) Surface expression of T cell exhaustion markers (PD-1, PD-L1, LAG-3, and TIM-3) was analyzed on day 12. The data shown are representative FACS plots (d) and the mean fluorescence

intensity of each marker in CD8⁺ T cell population (e, n=7 different donor samples, paired *t*-test). (f) SGC0946- or DMSO-treated T cells were restimulated by aAPC/mOKT3 on day 12. The fold expansion of the restimulated T cells were calculated at the indicated time points (n=4 technical replicates, unpaired *t*-test). Similar results were reproduced with a different donor sample. Horizontal lines indicate the means ± s.d. ns, not significant.

Comment #4-1.

The CAR studies are unclear and the claims with regards to CARs very broad. The CAR is not described (signaling elements, sequence). It is unclear whether the reported findings would hold up with a different CAR design.

Response to comment #4-1.

In the original manuscript, we used an anti-CD19 CAR construct containing the CD28 and 4-1BB signaling domains. We have included this information in the Methods section and presented the amino acid sequences of the signaling domains in Supplementary Table 4. In the revised manuscript, we have also tested a CAR construct containing a 4-1BB domain alone. While the presence of a CD28 signaling domain augments initial expansion of CAR-T cells upon encountering an antigen, it also accelerates T cell exhaustion and negatively affects long-term T cell persistence (Ghosh et al, Nat Med 2017).

BB-z CAR-T cells efficiently eradicated leukemia cells regardless of ectopic expression of DUSP6 (Supplementary Fig. 25b,c). Similar to 28-BB-z CAR-T cells, control but not DUSP6-overexpressing BB-z CAR-T cells subsequently induced xenogeneic lethal GVHD (Supplementary Fig. 25d-f).

* Page 13, line 10: Finally, we investigated the impact of DUSP6 overexpression on antitumor T cell responses and the development of GVHD. Human CD3⁺ T cells were transduced with an anti-CD19 chimeric antigen receptor (CAR) that had CD28 and 4-1BB cytoplasmic domains (28-BB-z)⁴¹.

* Page 20, line 22: Anti-CD19 CAR constructs (FMC63 anti-CD19 scFv linked with CD28, 4-1BB and CD3z, or CD8α, 4-1BB and CD3z) were also inserted into the pMX vector and linked to ΔNGFR or ΔNGFR and DUSP6 by a Furin-SGSG-P2A sequence. The amino acid sequences of the CAR signaling domains are provided in Supplementary Table 4.

* Page 14, Line 12: Similar results were found when T cells were transduced with a different design of the CAR construct that had a 4-1BB cytoplasmic domain alone (BB-z). While both control and DUSP6-overexpressing BB-z CAR-T cells controlled leukemia progression, only control CAR-T cells induced progressive weight loss in treated mice (Supplementary Fig. 25a-f).

* Supplementary Fig. 25:

Supplementary Fig. 25. Treatment of the CD19⁺ acute lymphoblastic leukemia cell line NALM-6 with anti-CD19 BB-z CAR-engineered T cells with or without ectopic expression of DUSP6. (a) CD3⁺ T cells were retrovirally transduced with an anti-CD19 CAR gene that had a 4-1BB signaling domain (BB-z) and ΔNGFR, or BB-z CAR, DUSP6 and ΔNGFR linked to a Furin-SGSG-P2A sequence. ΔNGFR⁺ cells were magnetically isolated and infused into NALM6-GL-bearing NSG mice using the same protocol described in Figure 6a. (b, c) Leukemia progression was monitored by *in vivo* bioluminescent imaging of the luciferase activity. The data shown are the total photon counts (b) and images of individual mice (c) at the indicated time points. (d) The frequencies of human CD45⁺ CD4⁺/CD8⁺ CAR-T cells in the peripheral blood (n=6 mice, unpaired *t*-test). (e) Sequential monitoring of body weight relative to the weight on day 0. (f) Kaplan-Meier analysis for overall survival after NALM6-GL infusion (n=6 mice, log-rank test). * P<0.05, ** P<0.01.

Comment #4-2.

The claim that low affinity TCR signaling would be impeded, but not CAR activity, is interesting but unsubstantiated. Clonal studies, perhaps including genetic analyses, would be needed to support this hypothesis. An *in vivo* experiment demonstrating preserved CAR function while preventing GVHD would provide the most direct functional assessment of this important claim.

Response to comment #4-2.

Thank you for your important comment. We have additionally performed the following *in vivo* experiments to show that DUSP6 overexpression preserves CAR functions while preventing xenogeneic GVHD.

First, we compared the *in vivo* functions of control and DUSP6-CAR-T cells in more detail (Supplementary Fig. 22). CAR-T cells were labelled with CFSE and infused into NALM6-bearing or tumor-free mice to assess the antigen-driven proliferation of CAR-T cells. As shown in Supplementary Fig. 22, b and c, both control and DUSP6-CAR T cells showed similar CFSE dilutions in the presence of NALM-6. We also measured IFN- γ secretion by control and DUSP6-CAR-T cells *in vivo* and found that both CAR-T cells similarly secreted IFN- γ in an antigen-dependent manner.

Control CAR-T but not DUSP6 CAR-T cells expanded after leukemia eradication at 3-5 weeks after T cell infusion (Fig. 6d and Supplementary Fig. 25d). To exclude the possibility that DUSP6-CAR T cells become more exhausted after antigen exposure and lose proliferative capacity independent of xenogeneic T cell responses, we repeatedly infused NALM6-GL into NSG mice after CAR-T cell infusion (Supplementary Fig. 24a). When analyzed in peripheral blood and the spleen, no significant differences in persistence were observed between control

and DUSP6 CAR-T cells, and both CAR-T cells efficiently eradicated the rechallenged NALM6-GL (Supplementary Fig. 24, b-d). These results confirmed that DUSP6 CAR-T cells maintained persistence capacity at the same level as control CAR-T cells in the presence of antigen.

Finally, we performed an experiment to assess whether endogenous TCR signaling is indeed required for the expansion of CAR-T cells after leukemia eradication. To achieve this goal, endogenous TCR expression in CAR-T cells was ablated by targeting the TCR α constant region using the CRISPR/Cas9 system. Expression of CD3 was deficient in approximately 30% of CAR-T cells. NALM6-bearing mice were infused with a mixed population of control and DUSP6-CAR-T cells with or without expression of endogenous TCR, and the relative abundance of each T cell population was monitored at multiple time points (Fig. 6i). Control and DUSP6-CAR-T cells were discriminated by the expression of EGFP or Δ NGFR, respectively. While each population showed a similar frequency on day 2 after CAR-T cell infusion, only control CAR-T cells expressing endogenous TCR progressively expanded in the peripheral blood (Figure 6, j and k). These results indicate that persisting CAR-T cells expand through endogenous TCR signaling in tumor-eradicated NSG mice, which is significantly attenuated by DUSP6 overexpression.

Analysis of GVHD development using T cell clones is technically difficult to perform. In a xenogeneic model using NSG mice, GVHD is primarily induced by the CD45RA⁺ CD62L⁺ CCR7⁺ T cell population (Cieri et al, Blood 2013; Gattinoni et al, Nat Med 2011). Since this less-differentiated T cell population is progressively lost during *in vitro* T cell expansion, the culture duration must be minimized to study the development of GVHD. The expanded T cell clones are fully differentiated into a CD45RO⁺ CD62L⁻ CCR7⁻ effector memory phenotype and poorly engraft in NSG mice (Chandran et al, Cancer Res 2015; Hsu et al, Blood 2007).

* Page 13, Line 18: **Next, we investigated the *in vivo* attributes of CAR-T cells with or without overexpression of DUSP6. When CFSE-labelled CAR-T cells were transplanted into NALM6-bearing or tumor-free NSG mice, both control and DUSP6-CAR⁺ T cells proliferated with similar efficiencies in an antigen-dependent manner (Supplementary Fig. 22a-c). There**

was no significant difference in IFN- γ secretion between control and DUSP6-CAR-T cells *in vivo* (Supplementary Fig. 22d).

* Supplementary Fig. 22:

Supplementary Fig. 22. DUSP6 overexpression in CAR-T cells do not impair CAR-T cell functions *in vivo*.

(a) NALM6-bearing or tumor-free mice were treated with 6 million CFSE-labelled 28-BB-z CAR-T cells with or without overexpression of DUSP6 (day 0). Mice were sacrificed on day 7, and analyzed for CFSE dilution of CAR-T cells within the spleen. (b, c) Representative FACS plots analyzing the CFSE dilution (b) and mean fluorescence intensity of CFSE in the $\Delta\text{NGFR}^+ \text{CD4}^+/\text{CD8}^+$ T cell population (c, n=6 mice for each condition, ordinary one-way ANOVA with Tukey's multiple comparisons test). (d) Serum was collected from NALM6-bearing or tumor-free mice at day 3 after CAR-T cell infusion. The serum concentration of IFN- γ was measured by ELISA (n=6 mice for each, unpaired *t*-test). ** P<0.01. ns, not significant; nd, not detected. In c and d, Horizontal lines denote the means \pm s.d.

* Page 14, Line 9: Importantly, DUSP6-CAR-T cells showed similar persistence compared with control CAR-T cells in mice rechallenged with NALM6-GL and prevented leukemia progression (Supplementary Fig. 24a-d). These results suggest that DUSP6 overexpression in CAR-T cells does not impair *in vivo* CAR-T cell proliferation in the presence of the target antigen.

* Supplementary Fig. 24:

Supplementary Fig. 24. DUSP6 overexpression in CAR-T cells does not compromise antitumor effects upon tumor rechallenge *in vivo*.

(a) NALM6-bearing leukemia mice were treated with 6 million CAR-T cells with or without overexpression of DUSP6 (day 0). NALM6-GL was reinjected into the mice on days 6 and 12, and the persistence of CAR-T cells and leukemia progression were analyzed. (b) The persistence of CAR-T cells in the peripheral blood was analyzed on days 6, 12 and 18 (n=6 mice for each group, unpaired *t*-test). (c, d) On day 18, mice were sacrificed and analyzed for infiltration of NALM6-GL and the persistence of CAR-T cells in the spleen. The data shown are representative FACS plots analyzing the frequency of GFP⁺ leukemia cells (c,

n=4 mice for “No T cell control” group, n=6 mice each for CAR-T cell treatment group) and the absolute number of the persisting CAR-T cells in the spleen (n=6, unpaired *t*-test). ns, not significant. Horizontal lines indicate the means \pm s.d.

* Page 14, Line 17: To confirm that the expansion of CAR-T cells and development of xenogeneic GVHD after leukemia eradication was induced by endogenous TCR expressed in CAR-T cells, we ablated TCR expression by targeting the TCR alpha constant region with the CRISPR/Cas9 system. The knockout efficiency was approximately 30% in both control and DUSP6-CAR-T cells (Supplementary Fig. 26a). Control EGFP⁺ CAR-T and DUSP6- Δ ANGFR⁺ CAR-T cells with or without expression of CD3 were coinjected into NALM6-bearing mice (Fig. 6i). We confirmed the comparable engraftment of each CAR-T cell population two days after transplantation (Fig. 6j; Supplementary Fig. 26b). However, only EGFP⁺ TCR⁺ CAR-T cells showed expansion at later time points (Fig. 6k). These results suggest that endogenous TCR signaling is required for CAR-T cell expansion in tumor-eradicated NSG mice through xenogeneic T cell responses, which is attenuated by overexpression of DUSP6.

* Fig. 6, i-k:

Fig. 6. (i-k) CD3⁺ T cells were retrovirally transduced with EGFP-P2A-BB-z CAR or Δ ANGFR-P2A-BB-z-P2A-DUSP6 CAR, and endogenous TCR expression was ablated using CRISPR/Cas9. NALM6-bearing mice were coinjected with CD3^{+/−} control and DUSP6-CAR-T cells (4 million T cells per mouse). The persistence of CAR-T cells was monitored in peripheral blood. The frequency of each CAR-T cell population among peripheral blood mononuclear cells (j) or CAR-T cells (k) is shown (n=6 mice, repeated

measures one-way ANOVA with Tukey's multiple comparisons test). Representative data of two experiments. ns, not significant. Horizontal lines indicate the means \pm s.d.

Comment #5.

As the effect of SGC0946 is transient and reversible after 2 weeks, the authors should discuss whether they consider a genetic or chemical approach to be the most clinically relevant.

Response to comment #5

Thank you for your comment. We have discussed clinical relevance of this study in the Discussion section as shown below.

* Page 18, line 5: Currently available DOT1L inhibitors have a short half-life in plasma and therefore require continuous infusion for *in vivo* studies⁴⁸. One of the inhibitors, EPZ-5676, has been tested in a phase I study for acute myeloid leukemia patients with rearrangements of the MLL gene, in which the inhibitor is administered as a 28-day continuous intravenous infusion (NCT01684150). In adoptive immunotherapy, the persistence of infused antitumor T cells is highly variable between studies. In a recent clinical trial using CD19 CAR-T cells with a 4-1BB costimulatory domain, CAR-T cells were detected in peripheral blood by flow cytometry up to 11 months⁴. As shown in our study, stable transduction of DUSP6 in CAR- or TCR-engineered antitumor T cells is a feasible alternative for long-term prevention of GVHD. Ablation of endogenous TCRs using the CRISPR/Cas9 system is another promising approach. However, an optimal protocol to efficiently introduce Cas9/sgRNAs into T cells with minimal toxicity remains to be established.

Reviewer #2's comments:

Comment #1.

Among epigenetic inhibitors used in the screen, there are several other active compounds (see for example Figure 1c). Are these compounds upregulating DUSP6 or likely work through different mechanism?

Response to comment #1

We investigated DUSP6 expression in T cells treated with GSK-J4, GSK-LSD1, UNC1215, JQ1, NI-57, and LAQ824. These compounds decreased either CD69 expression (GSK-LSD1, UNC1215 and LAQ824) or the CFSE dilution (GSK-J4, JQ1, and NI-57) in screening experiments (Fig. 1). Although GSK343, PFI-3, I-BRD9 and decitabine decreased cellular division in CD4⁺ or CD8⁺ T cells, they also attenuated T cell division in the absence of allogeneic cells (Supplementary Fig. 3).

As shown in Supplementary Fig. 15, none of the reagents significantly altered DUSP6 expression. Of note, we selected DOT1L as a promising target because only SGC0946 significantly decreased both CD69 expression and CFSE dilution in the screening experiment.

* Page 10, Line 20: **Although we also analyzed the DUSP6 expression levels in T cells treated with other epigenetic chemical probes that modulated allogeneic T cell responses (Fig. 1), none of the inhibitors affected DUSP6 expression (Supplementary Fig. 15a,b).**

* Supplementary Fig. 15:

Supplementary Fig. 15. Effects of different epigenetic chemical probes on DUSP6 expression levels. (a) CD3⁺ T cells were treated with the indicated chemical probes for 14 days. The DUSP6 protein levels were analyzed by immunoblotting. Representative blots of three experiments (a) and the quantified DUSP6 protein levels normalized to β -actin are shown (b, n=3, repeated measures one-way ANOVA with Tukey's multiple comparisons test). Horizontal lines indicate the means \pm s.d. ns, not significant.

Comment #2.

It is not clear how DUSP6 was identified as main target gene of DOT1L inhibitor. As described in the text, DUSP6 was selected as one of the genes affected by treatment of MLL-leukemia cell lines with DOT1L inhibitor. This sounds like semi-random guess and it is very likely that expression of many other genes have been affected by treatment with DOT1L inhibitor. More systematic analysis of gene expression by performing RNA-seq is needed in order to better understand the mechanism.

Response to comment #2

Thank you for your important comment. We performed RNA-sequencing analysis in CD8⁺ T cells treated with SGC0946 or DMSO (control) to compare their gene expression profiles (three different donor-derived samples per condition). We extracted 205 genes that showed significant changes in FPKM values ($P < 0.01$, $FDR < 0.1$ and fold change > 1.5 or < 0.66) between SGC0946- and DMSO-treated CD8⁺ T cells. DUSP6 expression was significantly increased in SGC0946-treated T cells (Supplementary Table 2).

As another approach, we compared multiple phosphoproteins related to T cell activation between SGC0946- and DMSO-T cells. Since the effect of SGC0946 is evident in the context of low-avidity T cell stimulation, T cells were transduced with low-affinity A2/MART-1 TCR (clone 413) and stimulated with the aAPC/A2 loaded with MART1₂₇₋₃₅ peptide. As shown in Fig. 4a, phosphorylation of ERK, but not p38 MAPK, Akt and CD3 ζ , was significantly decreased in SGC0946-treated T cells. We also analyzed JAK-STAT pathway activation upon treatment with cytokines (IL-21 for STAT3 and IL-2 for STAT5). SGC0946 treatment did not affect the phosphorylation levels of STAT3 or STAT5 (Supplementary Fig. 12). Based on these results, we searched for MAPK pathway-related genes among the list of differentially expressed genes and identified four genes (*DUSP2*, *DUSP6*, *RRAS2* and *FAS*). The MAPK pathway gene set, which includes genes related to the ERK, p38 MAPK and JNK pathways, was retrieved from the KEGG pathway database. Among these genes, we focused on *DUSP6* as a specific regulator of ERK phosphorylation. Although *DUSP2* also targets ERK, it also dephosphorylates p38 MAPK.

To further evaluate the impact of increased DUSP6 expression in SGC0946-treated T cells in allogeneic responses, we transduced SGC0946-treated T cells with a dominant-negative mutant of

DUSP6 C293S and analyzed the ERK phosphorylation and allogeneic responses in comparison to control Δ NGFR-transduced T cells treated with DMSO or SGC0946. As shown in Supplementary Fig. 14d, e, ectopic expression of DUSP6 C293S abrogated the suppressive effect of SGC0946 on ERK phosphorylation and allogeneic T cell proliferation. These data suggest that increased expression of DUSP6 has a predominant, if not absolute, role in the suppression of allogeneic T cell responses in SGC0946-treated T cells.

* Fig. 4a,b:

Fig. 4. (a) CD8⁺ T cells transduced with a low-affinity A2/MART1 T cell receptor (clone 413) and Δ NGFR were treated with SGC0946 or DMSO for 14 days and stimulated with the aAPC/A2 loaded with A2/MART1₂₇₋₃₅ peptide. The mean fluorescence intensity (MFI) of phospho-CD3z, ERK, p38 MAPK and Akt in CD8⁺ Δ NGFR⁺ T cells was analyzed (n=4 cultures, unpaired *t*-test). (b) Gene expression profiles of CD8⁺ T cells treated with SGC0946 or DMSO for 14 days were compared by RNA sequencing (n=3 different donor samples). MAPK-related genes were extracted from the differentially expressed genes.

* Supplementary Fig. 14d, e:

Supplementary Fig. 14. (d,e) CD3⁺ T cells were retrovirally transduced with a dominant-negative mutant of DUSP6 (C293S)-IRES-ΔNGFR or control ΔNGFR and treated with SGC0946 or DMSO. (d) ERK phosphorylation was analyzed 70 minutes after stimulation with PMA/ionomycin, and MFI in the CD8⁺ ΔNGFR⁺ T cell population was calculated (n=4 cultures, ordinary one-way ANOVA with Tukey's multiple comparisons test). (e) The indicated T cells were cocultured with allogeneic PBMC after being labeled with CFSE. The frequency of dividing cells in the ΔNGFR⁺ T cell population was analyzed 4 days later (n=4 cultures, ordinary one-way ANOVA with Tukey's multiple comparisons test for each T cell population). In e-e, Horizontal lines denote the means ± s.d. ns, not significant.

* Page 9, Line 6: To investigate how DOT1L inhibition modulates the T cell activation threshold, we explored the effect of SGC0946 treatment on the activation of multiple proximal and distal signaling pathways in low-avidity T cells transduced with the A2/MART1 TCR clone 413. Interestingly, phospho-ERK, but not the other phosphoproteins (CD3z, p38 MAPK, Akt), was significantly downregulated in SGC0946-treated T cells compared with the control (Fig. 4a and Supplementary Fig. 12a). Although we also analyzed JAK-STAT signaling pathway activation after treatment with cytokines, there were no significant differences in the STAT3 or STAT5 phosphorylation levels between SGC0946- and DMSO-treated T cells (Supplementary Fig. 12b,c). We then compared whole transcriptomes between SGC0946- and DMSO-treated CD8⁺ T cells by RNA sequencing analysis. Overall, we extracted 205 genes that were differentially expressed between the DOT1L-inhibited and control T cells (P<0.01, FDR [false discovery rate] <0.1, and fold change of FPKM [fragments per kilobase of transcript per million mapped reads] >1.5 or <0.66; Supplementary Table 2). Among the gene list, we identified 4 genes related to the MAPK signaling pathway (Fig. 4b; the MAPK signaling pathway genes were extracted from the KEGG pathway database). Intriguingly, the genes encoding phosphatases (*DUSP2* and

DUSP6) were significantly upregulated in SGC0946-treated T cells. While DUSP2 is involved in the dephosphorylation of both p38 MAPK and ERK, DUSP6 specifically targets ERK^{31, 32}.

* Page 10, Line 13: Conversely, ectopic expression of a dominant-negative form of DUSP6/C293S in SGC0946-treated T cells reverted the ERK phosphorylation levels and allogeneic responses to those of the control T cells, suggesting that the increased expression of DUSP6 played a predominant role in the suppression of allogeneic responses in SGC0946-treated T cells (Supplementary Fig. 14d,e).

Comment #3.

Furthermore, is the DUSP6 protein level achieved upon DUSP6 overexpression in T-cells (experiments in Figure 6) comparable to DUSP6 level upon treatment with DOT1L inhibitor?

Response to comment #3

We evaluated the protein levels of DUSP6 in DMSO- or SGC0946-treated T cells and in T cells retrovirally transduced with DUSP6. As shown in Supplementary Fig. 14, a and b, retroviral transduction of DUSP6 resulted in significantly higher expression levels compared with SGC0946 treatment.

We compared the kinetics of ERK dephosphorylation and the allogeneic responses between SGC0946-treated and DUSP6-transduced T cells. As shown in Fig. 4f, g and Supplementary Fig. 14c, there were no significant differences in the ERK phosphorylation levels and allogeneic responses between SGC0946-treated and DUSP6-transduced T cells. These results suggest that the increased DUSP6 expression achieved by SGC0946 is sufficient to promote ERK dephosphorylation and suppress allogeneic T cell responses as efficiently as retroviral overexpression.

* Page 10, Line 10: Although retroviral transduction resulted in significantly higher DUSP6 protein levels than SGC0946 treatment, there were no significant differences in the efficiency of ERK dephosphorylation or suppression of allogeneic responses between DUSP6-transduced and SGC0946-treated T cells.

* Supplementary Fig. 14a-c:

Supplementary Fig. 14. (a) $CD3^+$ T cells were retrovirally transduced with DUSP6-IRES- Δ NGFR or control Δ NGFR. Control Δ NGFR-transduced cells were treated with DMSO or SGC0946. Δ NGFR $^+$ cells were isolated and subjected to immunoblotting analysis with an anti-DUSP6 antibody. Representative blots of four experiments (a) and the quantified DUSP6 protein levels normalized to b-actin are shown (b, n=4, paired *t*-test). (c) ERK phosphorylation after stimulation with PMA/ionomycin was evaluated in T cells transduced with control Δ NGFR or DUSP6. Control Δ NGFR-transduced cells were treated with DMSO or SGC0946. The mean fluorescence intensity (MFI) in the $CD8^+$ Δ NGFR $^+$ T cell population is shown (n=4 cultures, ordinary one-way ANOVA with Tukey's multiple comparisons test for each time point).

* Fig. 4f, g:

Fig. 4. (f,g) CD3⁺ T cells were transduced with DUSP6-IRES-ΔNGFR or control ΔNGFR and treated with SGC0946 or DMSO. T cells were labeled with CFSE and cocultured with allogeneic PBMCs. The frequencies of dividing cells (f) and CD69⁺ cells (g) in the ΔNGFR⁺ CD8⁺ T cell population were analyzed (n=4 cultures, ordinary one-way ANOVA with Tukey's multiple comparisons test).

Comment #4.

Mechanistic link between DOT1L inhibition, downregulation of miR-181a and upregulation of DUSP6 is weak and more data are needed to convincingly validate this mechanism. As presented in Supplementary Fig 4, DOT1L inhibitor completely abolished H3K79me2 and reduction of H3K79me2 level at miR-181a transcription start site is expected. Can you knockdown miR-181a and analyze DUSP6 level? More data are needed to explain how treatment with DOT1L inhibitor upregulates DUSP6.

Response to comment #4

Thank you for your comment. We transiently transfected T cells with a commercially validated miR-181a inhibitor (mirVanaTM miRNA Inhibitor; ThermoFisher Scientific) and analyzed DUSP6 expression by immunoblotting. As shown in Supplementary Fig. 16, miR-181a inhibition significantly increased DUSP6 expression in cultured T cells.

We also treated SGC0946- or DMSO-treated T cells with a miR-181a mimic (mirVanaTM miRNA Mimic; ThermoFisher Scientific) and analyzed DUSP6 expression. Consistent with the inhibitor experiment, transfection of the miR181a mimic significantly decreased DUSP6 expression and abrogated the effect of SGC0946 treatment (Fig. 4, k and l). The transfection protocol has been described in the Methods section.

In addition to miR-181a, previous studies have identified several other molecules that regulate the DUSP6 protein levels: ETS1, ETS2, PUM2 and ZFP36 (Ref. #37 and 38 in the revised

manuscript). We confirmed that there was no significant difference in the expression of these genes between SGC0946- and DMSO-T cells (Supplementary Fig. 17). These results collectively suggest that downregulation of miR-181a plays a predominant role in the increased DUSP6 expression in SGC0946-treated T cells.

* Page 10, Line 23: **We confirmed that inhibition of miR-181a in cultured T cells elevated DUSP6 expression (Supplementary Fig. 16a,b).**

* Supplementary Fig. 16:

Supplementary Fig. 16. DUSP6 expression upon inhibition of miR-181a. (a, b) CD3⁺ T cells were transfected with control or a miR-181a inhibitor. DUSP6 expression was analyzed by immunoblotting 48 hours after transfection. Representative blots of five experiments (a) and the quantified DUSP6 protein levels normalized to β -actin are shown (b, n=5, one sample *t*-test). Horizontal lines indicate the means \pm s.d.

* Page 11, Line 3: **Moreover, treatment with a miR-181a mimic decreased DUSP6 expression in SGC0946-treated T cells to the same level as in control T cells (Fig. 4k,l).**

* Page 29, Line 23:

Transfection with a miRNA inhibitor or mimic

To inhibit or upregulate the function of miR-181a, 2 million T cells were transfected with 200 pmoles of the mirVana miR-181a inhibitor, mimic or negative control (Thermo Fisher Scientific) with NucleofectorTM 2b device (Lonza). The T007 program was used for electroporation. Total cell lysates were extracted from transfected T cells after 48 hours and used for immunoblotting.

* Fig. 4, k and l:

Fig. 4. (k,l) SGC0946- or DMSO-treated T cells were transfected with a miR-181a mimic. DUSP6 expression was analyzed by immunoblotting. Representative blots (k) and quantified protein levels are shown (l, n=4, repeated measures one-way ANOVA with Tukey's multiple comparisons test).

* Page 11, Line 7: We further explored several other genes that have previously been reported to regulate DUSP6 expression (*ETS1*, *ETS2*, *PUM2* and *ZFP36*)^{37, 38}, none of which showed a significant difference in expression levels between control and SGC0946-treated T cells (Supplementary Fig. 17). These results suggest that suppression of miR-181a expression plays a predominant role in DUSP6 upregulation by SGC0946 treatment.

* Supplementary Fig. 17:

Supplementary Fig. 17. Expression analysis of genes that regulate DUSP6 expression. CD3⁺ T cells were treated with SGC0946 or DMSO for 14 days. The expression levels of the indicated genes in the SGC0946-treated T cells relative to the DMSO-treated control T cells were analyzed by qPCR (n=5 different samples, one-sample *t*-test). Horizontal lines indicate the mean values. ns, not significant.

Comment #5.

It is not clear how long was the treatment with DOT1L inhibitor in gene expression studies in activated T-cells (see page 10, Fig 5). Was the treatment long enough to decrease H3K79me2?

Response to comment #5

In Fig. 5, T cells were treated with SGC0946 or DMSO for 10 days before stimulation. We confirmed that the H3K79me2 levels were substantially decreased in SGC0946-treated T cells (Supplementary Fig. 18). We have described the treatment protocol in more detail in the Methods section.

* Supplementary Fig. 18:

Supplementary Fig. 18. Analysis of the H3K79me2 levels in DMSO- or SGC0946-treated CD8⁺ T cells. CD8⁺ T cells derived from three healthy donors were transduced with the DMF5 or clone 413 A2/MART1 TCR and treated with SGC0946 or DMSO (control) for 10 days. Repression of the H3K79me2 levels in SGC0946-treated T cells was confirmed by immunoblotting.

* Page 11, Line 23: CD8⁺ T cells were transduced with the DMF5 or clone 413 TCR and cultured with or without SGC0946 for 10 days to repress the H3K79me2 levels (Supplementary Fig. 18).

* Page 26, Line 21: For microarray gene expression analysis of activated T cells, CD8⁺ T cells derived from three different donors were individually transduced with DMF5 or clone 413 TCR and cultured in the presence of SGC0946 or DMSO for 10 days.

REVIEWERS' COMMENTS:

Reviewer #1 (Remarks to the Author):

The title and messaging are exaggerated ("...enables the safe and effective use of allogeneic...") and not aligned with the findings. The authors state in the text that DOT1L inhibition "delays the development of GVHD" (p.9) or "attenuated the development of GVHD" (p.43). The strength of alloreactivity may be variable under different circumstances (genetic background, conditioning, microenvironment, T cell dose, etc) and conclusions should therefore be nuanced. "Suppresses the incidence of GVHD" (in the text and legends) should also be corrected.

Reviewer #2 (Remarks to the Author):

The manuscript by Kagoya et al describes link between pharmacologic inhibition of DOT1L and allogeneic T-cell response in preventing GVHD. The revised manuscript is significantly improved and authors addressed all my concerns by performing new experiments including RNA-seq, additional mechanistic studies linking DOT1L inhibition and miR-181a, providing additional experimental details and clarification of the findings. In summary, this is very well written manuscript that describes important finding, and may have translational implications in preventing GVHD. I recommend the manuscript for publication.

Reviewer #1's comments:

Comment #1.

The title and messaging are exaggerated (“...enables the safe and effective use of allogeneic...”) and not aligned with the findings. The authors state in the text that DOT1L inhibition “delays the development of GVHD” (p.9) or “attenuated the development of GVHD” (p.43). The strength of alloreactivity may be variable under different circumstances (genetic background, conditioning, microenvironment, T cell dose, etc) and conclusions should therefore be nuanced. “Suppresses the incidence of GVHD” (in the text and legends) should also be corrected.

Response to comment #1.

Thank you for the comment. We have corrected the title and text as shown below.

* Title: **DOT1L inhibition attenuates graft-versus-host disease by allogeneic T cells in adoptive immunotherapy models**

* Page 2, line 9 (Abstract): The inhibition of DOT1L or DUSP6 overexpression in T cells **attenuates** the development of graft-versus-host disease while retaining potent antitumor activity in xenogeneic and allogeneic adoptive immunotherapy models.

* Page 7, line 14 (subheading): **DOT1L inhibition attenuates graft-versus-host disease**

* Page 13, line 4 (subheading): **DUSP6 overexpression in antitumor T cells ameliorates GVHD**

* Page 41, line 21 (legend for Figure 3): DOT1L inhibition **delays** the incidence of GVHD in a xenogeneic model.